# A hemoprotein with a zinc-mirror heme site ties heme availability to carbon metabolism in cyanobacteria

Nicolas Grosjean [1,13], Estella F. Yee [2], Desigan Kumaran[1], Kriti Chopra [3], Macon Abernathy[4], Sandeep Biswas [5], James Byrnes[2], Dale F. Kreitler[2], Jan-Fang Cheng [6], Agnidipta Ghosh [7], Steven C. Almo[7], Masakazu Iwai[8,9], Krishna K. Niyogi [8,9,10], Himadri B. Pakrasi [5], Ritimukta Sarangi [4], Hubertus van Dam [11], Lin Yang [2], Ian K. Blaby [6,12] & Crysten E. Blaby-Haas [1,13,14] ✉

Heme has a critical role in the chemical framework of the cell as an essential protein cofactor and signaling molecule that controls diverse processes and molecular interactions. Using a phylogenomics-based approach and complementary structural techniques, we identify a family of dimeric hemoproteins comprising a domain of unknown function DUF2470. The heme iron is axially coordinated by two zinc-bound histidine residues, forming a distinct two-fold symmetric zinc-histidine-iron-histidine-zinc site. Together with structure-guided in vitro and in vivo experiments, we further demonstrate the existence of a functional link between heme binding by Dri1 (Domain related to iron 1, formerly ssr1698) and post-translational regulation of succinate dehydrogenase in the cyanobacterium *Synechocystis*, suggesting an iron-dependent regulatory link between photosynthesis and respiration. Given the ubiquity of proteins containing homologous domains and connections to heme metabolism across eukaryotes and prokaryotes, we propose that DRI (Domain Related to Iron; formerly DUF2470) functions at the molecular level as a heme-dependent regulatory domain.

Cyanobacteria are essential contributors to multiple biogeochemical cycles[1] with marine cyanobacteria being responsible for 25% of the ocean net primary productivity[2]. To perform these functions, cyanobacteria have high iron requirements often exceeding other bacteria[3] and must ensure that the biosynthesis of iron cofactors[4] (e.g., heme) matches demand, such as for the synthesis of photosynthetic and respiratory complexes. In contrast to eukaryotic phototrophs, where respiratory and photosynthetic electron transfer chains are separated

[1]Biology Department, Brookhaven National Laboratory, Upton, NY, USA. [2]National Synchrotron Light Source II, Brookhaven National Laboratory, Upton, NY, USA. [3]Computational Science Initiative, Brookhaven National Laboratory, Upton, NY, USA. [4]Stanford Synchrotron Radiation Lightsource, SLAC National Accelerator Laboratory, Menlo Park, CA, USA. [5]Department of Biology, Washington University, St. Louis, MO, USA. [6]US Department of Energy Joint Genome Institute, Lawrence Berkeley National Laboratory, Berkeley, CA, USA. [7]Department of Biochemistry, Albert Einstein College of Medicine, Bronx, NY, USA. [8]Department of Plant and Microbial Biology, University of California, Berkeley, CA, USA. [9]Molecular Biophysics and Integrated Bioimaging Division, Lawrence Berkeley National Laboratory, Berkeley, CA, USA. [10]Howard Hughes Medical Institute, University of California, Berkeley, CA, USA. [11]Condensed Matter Physics and Materials Science Department, Brookhaven National Laboratory, Upton, NY, USA. [12]Environmental Genomics and Systems Biology Division, Lawrence Berkeley National Laboratory, Berkeley, CA, USA. [13]Present address: US Department of Energy Joint Genome Institute, Lawrence Berkeley National Laboratory, Berkeley, CA, USA. [14]Present address: The Molecular Foundry, Lawrence Berkeley National Laboratory, Berkeley, CA, USA. ✉e-mail: cblaby@lbl.gov

in distinct organelles, the two energetic processes are physically connected in cyanobacteria[5,6]. For instance, the well-characterized type I and type II NADH dehydrogenases (NDHs) and succinate dehydrogenase[7-9] (SDH) functionally contribute electrons to both electron transfer chains by participating in redox poising of the plastoquinone pool. Sharing iron-dependent electron carriers between the two electron transfer chains, however, adds an extra regulatory burden, whereby iron homeostasis, photosynthesis, and respiration need to be tightly coordinated.

Besides their vital role in electron transfer, iron cofactors such as heme are involved in multiple cellular functions. As an example, heme is a key signaling molecule that modulates cellular processes ranging from transcription to signal sensing and post-translational regulation of important proteins[10-12]. As such, regulatory mechanisms involving heme evolved in photosynthetic organisms to maintain optimal photosynthetic efficiency[13] while stymying oxidative stress and photo damage. In land plants, a heme-dependent mechanism evolved to control the abundance of glutamyl-tRNA reductase (GluTR1), a protein that initiates the first step of tetrapyrrole biosynthesis. Post-translational regulation of GluTR1 is essential to prevent excess production of heme and its intermediates[14,15]. This heme-dependent negative feedback regulation relies on the GluTR1-binding protein (GBP)[15], which upon binding to the regulatory domain of GluTR1, prevents GluTR1 degradation by Clp proteases[15]. Heme binding to GBP inhibits this interaction, leading to the degradation of GluTR1. GBP is composed of an N-terminal split-barrel domain (part of the pyridoxamine phosphate oxidase-like family) and a C-terminal domain of unknown function, DUF2470[16]. These two domains are also found in the HugZ-like protein family involved in heme degradation and iron acquisition in bacteria[17-19]. DUF2470 is associated with different molecular functions; this domain is thought to partially shield the heme-binding pocket of HugZ, whereas in GBP, DUF2470 is involved in the recognition of GluTR1. While heme binding to GBP has been demonstrated[15], the heme-bound form has yet to be structurally determined[20], and it is uncertain which of the two domains binds heme. Therefore, a consistent structure-function relationship is yet to be defined for DUF2470.

In this work, we use a phylogenomic-based approach to describe the large DUF2470-containing protein family. Based on this analysis, we hypothesize that DUF2470 is a heme-binding regulatory domain and have renamed DUF2470 to Domain Related to Iron (DRI). To test this hypothesis, we identify a cyanobacterial-specific protein clade where DRI is not fused to another domain and named this orthologous group of proteins Dri1. Dri1 is able to bind heme, and the heme-bound crystal structure revealed a Zn-mirror heme site. Structure-guided genetics and phenotypic analysis, using the genetic model *Synechocystis* sp. PCC 6803 (hereafter *Synechocystis*), support the involvement of Dri1 in the regulation of electron transfer specifically by altering the activity of succinate dehydrogenase (SDH) in a heme-dependent manner. The discovery of this family of DRI-containing proteins provides key molecular and genomic insights into the evolution of DRI as a heme-binding regulatory domain throughout different phyla and represents an unprecedented example of how heme can be coordinated to proteins.

## Results

### DUF2470 is a Domain Related to Iron (DRI) and is linked to heme homeostasis

We performed a phylogenomic analysis of DUF2470 to define its occurrence and genomic context. DUF2470 is encoded in the genomes of bacteria, archaea, and eukaryotes and is often fused to a pyridoxamine 5′-phosphate oxidase-like (PNPOx-like) domain in Pseudomonadota, Actinomycetota, and Viridiplantae (Fig. 1a). In Pseudomonadota, a relatively small subfamily of DUF2470-containing proteins (HugZ[17,18] and ChuZ[21]) have been shown to degrade *b*-type heme but are not related to

either canonical or IsdG-like heme oxygenases (HOs). In the crystal structures of HugZ, ChuZ, and MSMEG_6519[22], heme is bound at the PNPOx-like domain, while in ChuZ, a second surface-bound heme is coordinated to two conserved histidine residues in DUF2470 (Fig. 1b). The presence of this second heme-binding site is suggested to be conserved in HugZ and HugZ-like proteins[21]. We identified two distinct subfamilies of plant-specific homologs with the same domains as HugZ but the order is swapped (Fig. 1a, b). One subfamily includes the previously characterized GBP from *Arabidopsis thaliana*[15]. The other subfamily (HOZ) is phylogenetically distinct from GBP and, similar to the HugZ subfamily, is composed of heme-degradation proteins[23] (Fig. 1a). In fungi, DUF2470 is fused to a domain similar to transmembrane protein 254 (TMEM254), and in a small Pseudomonadota subfamily, DUF2470 is fused to a domain with similarity to a siderophore-interacting protein (PF04954) (Fig. 1a, b). Conserved gene neighborhood analysis reveals that genes encoding DUF2470-containing proteins are often next to genes encoding proteins putatively involved in iron homeostasis (Fig. 1a, Supplementary Fig. 1). This analysis suggests a conserved function of DUF2470 in iron and heme metabolism. Consequently, we renamed DUF2470 as Domain Related to Iron (DRI).

DRI is found as a single-domain protein in two phyla. In Actinomycetota, a distinct subfamily is either fused to a mammalian-type "canonical" heme oxygenase (HO) (PF01126) or, more often, is a single-domain protein encoded in a putative operon with a gene encoding HO (Fig. 1a, b), suggesting a strong functional link between these Actinomycetota DRI proteins and HO. In cyanobacteria, DRI is almost always found as a single-domain protein. The only exceptions are a homolog from *Coleofasciculaceae* cyanobacterium SM2_3_26 (N-terminal haemolysin-like domain (IPR001343)) and a HugZ-like protein from Cyanobacteria bacterium HKST-UBA02. While fusion proteins or conserved gene neighbors were not found to be informative for the single-domain cyanobacterial proteins, several protein-protein interactions with iron cofactor-dependent proteins were previously captured by high-throughput yeast two-hybrid (Y2H)[24].

### Dri1 contains a zinc-mirror heme-binding site

We heterologously expressed and purified the recombinant protein encoded by the *Synechocystis* gene *dri1* (formerly known as *ssr1698*) and tested the heme-binding ability of this small, soluble protein (96 amino acids, 10 kDa) (Fig. 2a). Dri1 with added heme contains 37.2 ± 0.3% of bound iron relative to protein, as measured by ICP-MS, suggesting that Dri1 could bind heme as an oligomer. Additionally, Dri1 with heme displays a characteristic Soret peak at 405 nm, weak signals at 500 nm and 540 nm in the Q-band region, and a charge transfer band at 625 nm, which corresponds to the formation of a Dri1-oxidized heme complex (Fig. 2a, b). Saturation heme-binding assays demonstrated a dissociation constant ($K_d$) of 0.15 μM, which is comparable to other hemoproteins with high heme-binding affinity[25]. Maximal heme-binding by Dri1 was achieved at pH values between 6.5 and 7.5 (Fig. 2c), indicative of heme-binding in the cytoplasm (neutral pH) rather than the thylakoid lumen (acidic pH). Indeed, a Dri1-eYFP fusion localizes to the cytosol in *Synechocystis* (Fig. 2d). Unlike HugZ and HOZ proteins, Dri1 does not display heme-degradation activity in the presence of an electron donor (Supplementary Fig. 2a), suggesting that the split-barrel domain (missing in Dri1) is responsible for heme-degradation by HugZ and HOZ proteins (Fig. 1b). Furthermore, we did not identify a homologous split-barrel domain protein-encoding gene in the *Synechocystis* genome, indicating that Dri1 does not associate with a split-barrel domain protein to form a two-subunit HugZ-like protein.

We determined the structures of Dri1 with and without heme to 2.35 Å and 3.00 Å resolution (Supplementary Table 1), respectively (Fig. 2e–h). Because of the potential structural flexibility and dynamics of the protein, we achieved a lower resolution with high Wilson B-factors (Supplementary Table 1) despite repeated efforts for data collection. Nonetheless, the quality and resolution of the data were

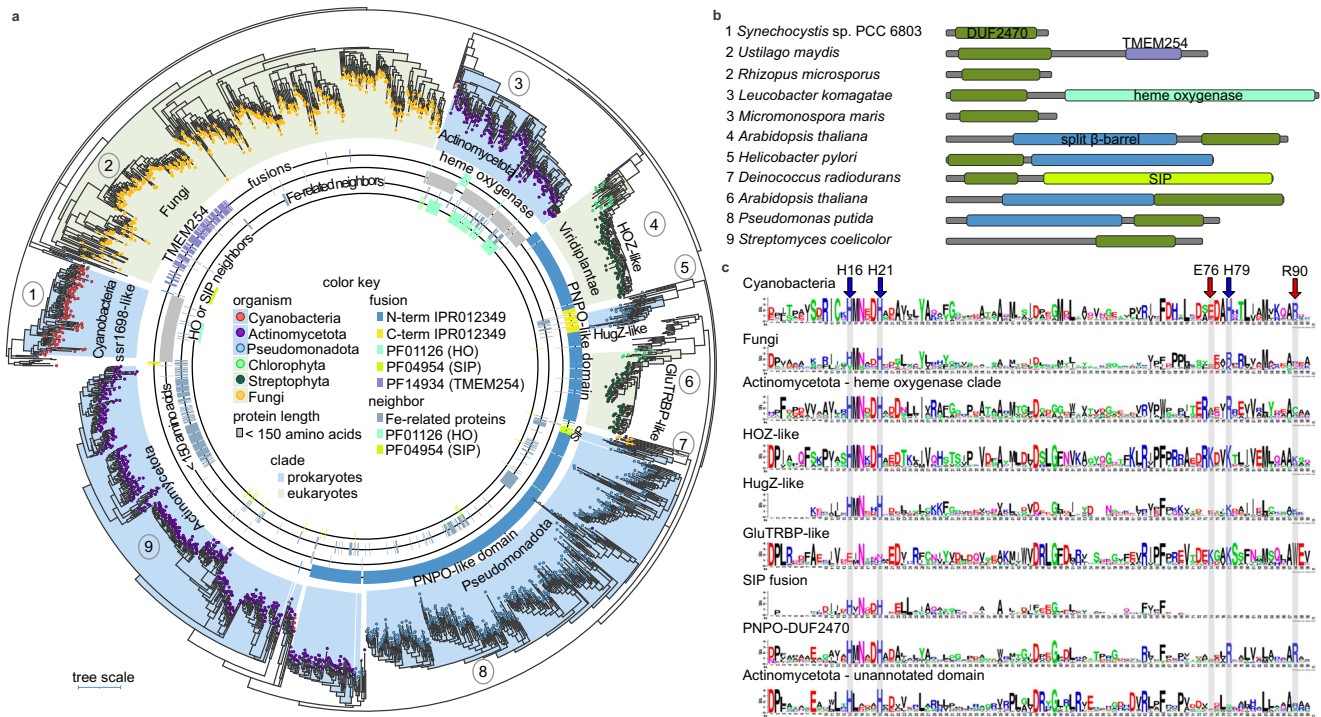

**Fig. 1 | DRI is linked to heme homeostasis. a** Phylogenetic tree of the DRI-containing protein family. Background color shading of clades is blue for prokaryotes and green for eukaryotes, while circles at the tips of leaves indicate phylum as given in the color key. Inner circles display information about the protein represented by each leaf according to the color key. The outermost circle indicates whether the protein contains TMEM254 (purple), SIP (light green), HO (teal), or the PNPO-like domain at either the N-terminus (blue) or C-terminus (yellow). The second outermost circle indicates whether the protein is less than 150 amino acids in length (grey). The third circle indicates whether a gene encoding a protein predicted to be involved in Fe homeostasis or Fe binding is found near the gene encoding the DUF2470-containg protein (greyish blue). The innermost circle indicates whether the gene encoding the DUF2470 protein is near a gene encoding either HO (teal) or SIP (light green). **b** Cartoon representation of the domain structure of selected DRI-containing proteins. Numbers to the left correspond to numbered clades in panel **a**. Coloring is used to indicate distinct and shared domains: DUF2470 (dark green), TMEM254 (purple), heme oxygenase (Pfam domain PF01126; teal), the PNPO-like split barrel domain (dark blue), and the SIP domain (light green). **c** HMMER alignment and WebLogo representation showing the conservation of DRI sequence in each clade. Source data are provided as Supplementary Data 1.

sufficient to shed light on the heme binding and residue interactions. The native structure devoid of heme is a single monomeric chain with a fold comparable to other DUF2470 structures in fusion proteins and comprises of three histidine residues (His16, His21, and His79; one on each alpha helix) that form a tetrahedral coordination site for a zinc ion (Fig. 2e–f). The heme-bound form is a homodimer consisting of two monomeric chains running in the same N-C direction. Although the homodimer maintains the same monomeric fold per chain, a two-fold-symmetric zinc-mirror site binds a single heme cofactor (Fig. 2g–h, Supplementary Fig. 3 for electron density maps). The heme iron is axially ligated at the interface of the dimer by His79 (Nε2; 2.04–2.13 Å) from each monomer. His79 (Nδ1; 2.11–2.14 Å) is in turn bound to a zinc ion that is additionally ligated by His16, His21, and Glu76. The heme propionate groups form a salt bridge with an Arg90 side chain from each monomer. Looking at the sequence conservation of these residues (Fig. 1c), we observe strict conservation of His16 and His21 (numeration based on the *Synechocystis* protein sequence) throughout different phyla, with the exception of GBP. However, Glu76, His79, and Arg90 residues are specific to cyanobacterial proteins and are not conserved in other DRI-containing proteins (Fig. 1c). This sequence analysis suggests that, although heme-binding is likely conserved, this mode of heme coordination is specific to Dri1 from cyanobacteria.

To provide more evidence for the structural models and analyze the structure of this protein in solution, size-exclusion chromatography coupled to small angle X-ray scattering (SEC-SAXS) was performed with and without heme (Fig. 3a–d, Supplementary Fig. 2b–d). Radius of gyration ($R_g$) values from Guinier analyses and molecular weight estimations confirm that Dri1 without heme remains

monomeric and heme-bound Dri1 is dimeric (Fig. 3d). SEC-SAXS shows that heme binding pushes the system primarily into the dimer state and no higher-order oligomers or unreacted monomers are present (Fig. 3a–d). Dimensionless Kratky plots further indicate that binding to heme decreases protein flexibility (Supplementary Fig. 2b). Electron density maps calculated by DENSS[26] yield consistent monomeric and dimeric reconstructions between datasets. To avoid overinterpretation[27,28], only the outermost contours (thresholds determined by approximate protein volumes) were considered in the comparisons, and scattering profiles were rebinned to a nonuniform-step $q$-grid to decrease noise in the low $q$ region and aid with data fitting (Fig. 3a).

Instead of relying on calculated electron densities, molecular dynamics (MD) ensembles were generated to model the possible solution-state structures. SAXS-driven MD calculations were run using the crystal structure as the initial state with additional flexible C-terminal tails from the remaining TEV cleavage site. The ensembles superpose well onto the DENSS electron density maps and confirm that the model fits are reasonable (Fig. 3e, Supplementary Fig. 4, Supplementary Table 2). The core of the dimer from the models agrees well with the crystal structure, indicating that the heme-binding region is relatively invariable.

To provide further support for the presence of the zinc-mirror heme site, we performed Fe K-edge XAS on apo-Dri1 +heme and Zn-Dri1 +heme (Fig. 3f–g, Supplementary Fig. 5). The EXAFS data support a six-coordinate Fe site in the protein with 6 Fe-N contributions at 2.04 Å (Supplementary Fig. 5c). Separation of these paths into two components (4 equatorial Fe-N paths from the heme ring and 2 Fe-N paths

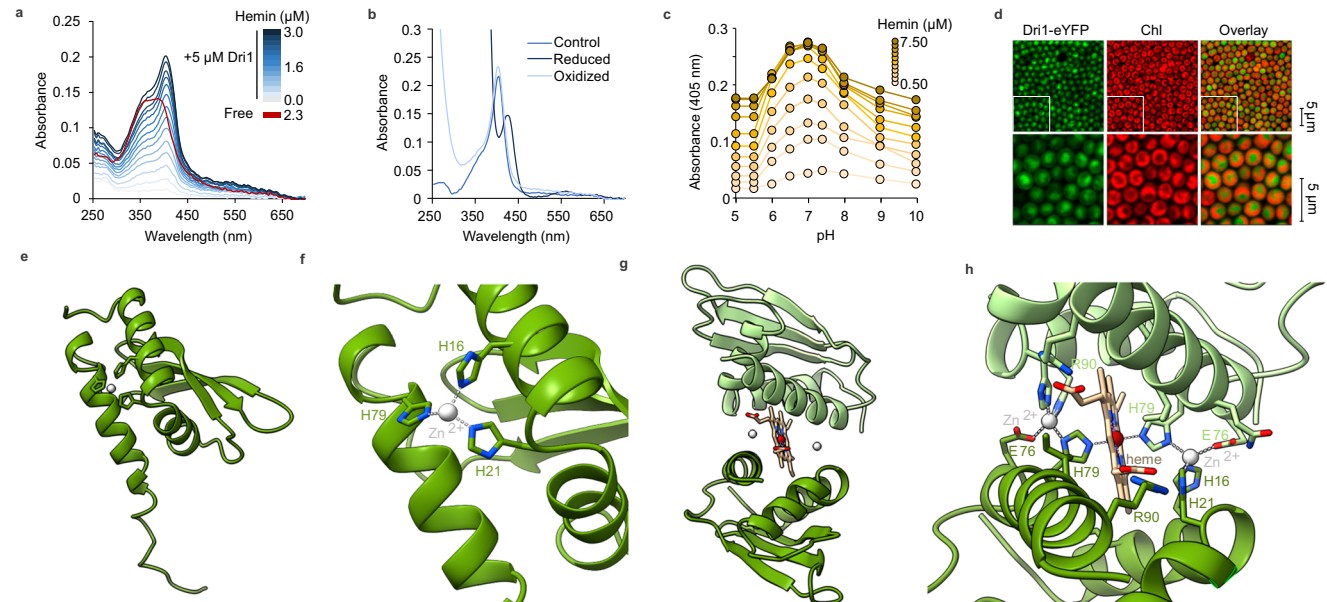

**Fig. 2 | Dri1 is a heme-binding protein with a zinc-mirror heme-binding site.**
**a** UV-Vis absorption spectra of heme binding by Dri1 show a characteristic Soret peak at 405 nm, indicating heme binding by Dri1. **b** UV-Vis absorption spectra of as-purified, ammonium persulfate-oxidized, and sodium dithionite-reduced forms of heme-bound Dri1. **c** Soret peak absorbance (405 nm) of heme-bound Dri1 at different pH and hemin concentrations. **d** Cytoplasmic localization of a Dri1-eYFP fusion in Synechocystis cells as observed by confocal microscopy. 'Chl' represents the chlorophyll autofluorescence used to localize thylakoids. A representative image of two individual experiments is shown. **e**–**f** Cartoon representations (green; ribbons for helices and arrows for β strands) of the monomeric Dri1 structure without heme. Side chains (in stick representation) that coordinate Zn (white sphere) are highlighted and labeled. **g**–**h** Cartoon representation of heme-bound (stick representation) Dri1 dimer (color coded in two distinct shades of green) (**g**) with focus on the Zn-mirror heme coordination site at the interface of the monomers (**h**). Dri1 side chains involved in heme coordination are shown in stick representation and labeled. Source data are provided as a Source Data file.

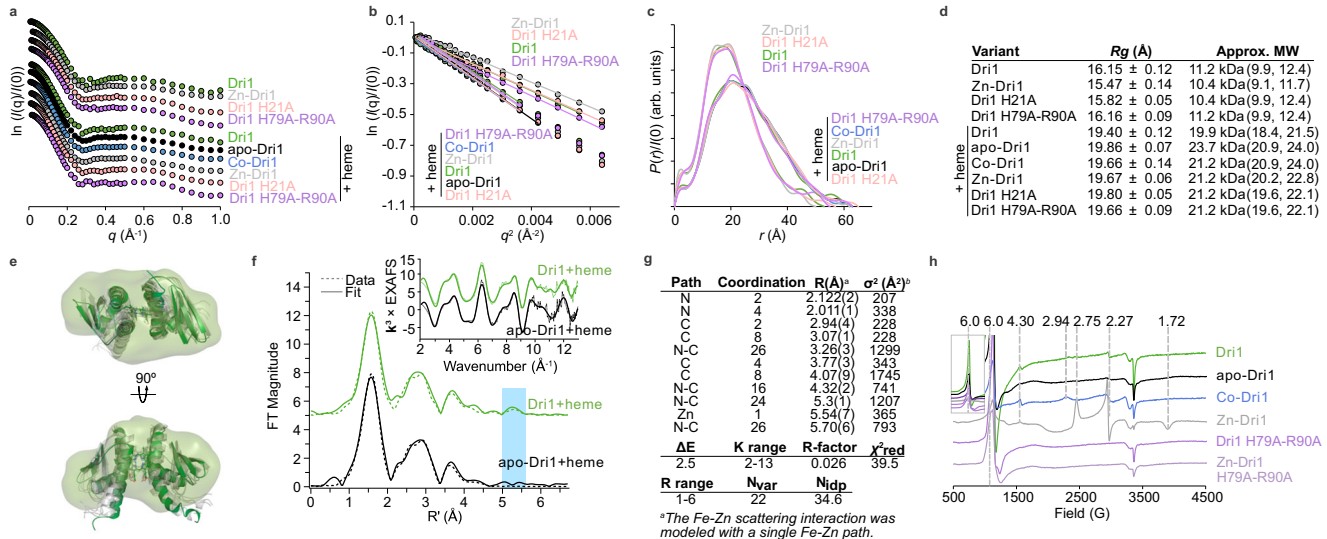

**Fig. 3 | Solution state structure of Dri1. a-b** Experimental SAXS scattering profiles (binned using a nonuniform-step q-grid and vertically offset for clarity), and corresponding Guinier plots (linear fits displayed up to $q_{max}R_g < 1.3$). **c** $P(r)$ distribution curves normalized to $I(0)$ for the same set of Dri1 samples to $q_{max} = 1.0 \text{ Å}^{-1}$. **d** $R_g$ values and approximate molecular weights of Dri1 samples calculated from Guinier analysis and by Bayesian inference with $q_{max} = 0.3 \text{ Å}^{-1}$, respectively. Confidence intervals are in parentheses. **e** SAXS-MD ensembles superimposed with DENSS envelopes calculated from SAXS data to $q_{max} = 1.0 \text{ Å}^{-1}$. **f** Iron K-edge EXAFS. Non-phase-corrected Fourier-transformed EXAFS of Dri1 and apo-Dri1. Note the Zn peak between 5 and 5.5 Å (indicated by a blue strip) in Dri1 absent in apo-Dri1. Inset: The corresponding $k^3$-weighted EXAFs. **g** Fe EXAFS least-squares fitting parameters for

Dri1.[a] The estimated standard deviations for distance are on the order of ± 0.02 Å. [b] Values of $\sigma^2$ have been multiplied by $10^5$. The value of $S_0^2$ was set to 1 for all fits. Coordination numbers have an error of ± 20%. **h** Heme-bound Dri1, apo-Dri1, Zn-Dri1, Co-Dri1, Dri1 His79Ala-Arg90Ala, and Zn-Dri1 His79Ala-Arg90Ala measured by X-band CW-EPR at 10 K. Strong signal intensities at $g = 6$ indicate high spin ($S = 5/2$) iron species. The addition of either $Co^{2+}$ or $Zn^{2+}$ significantly reduces the $g = 6$ signal intensity (inset) and leads to features at $g = 2.75$, 2.27, and 1.72, corresponding to low spin ($S = 1/2$), 6-coordinated iron. The $g = 4.3$ signal likely originates from non-heme ferric iron contamination. All intensities are normalized by protein concentration. Source data are provided as a Source Data file.

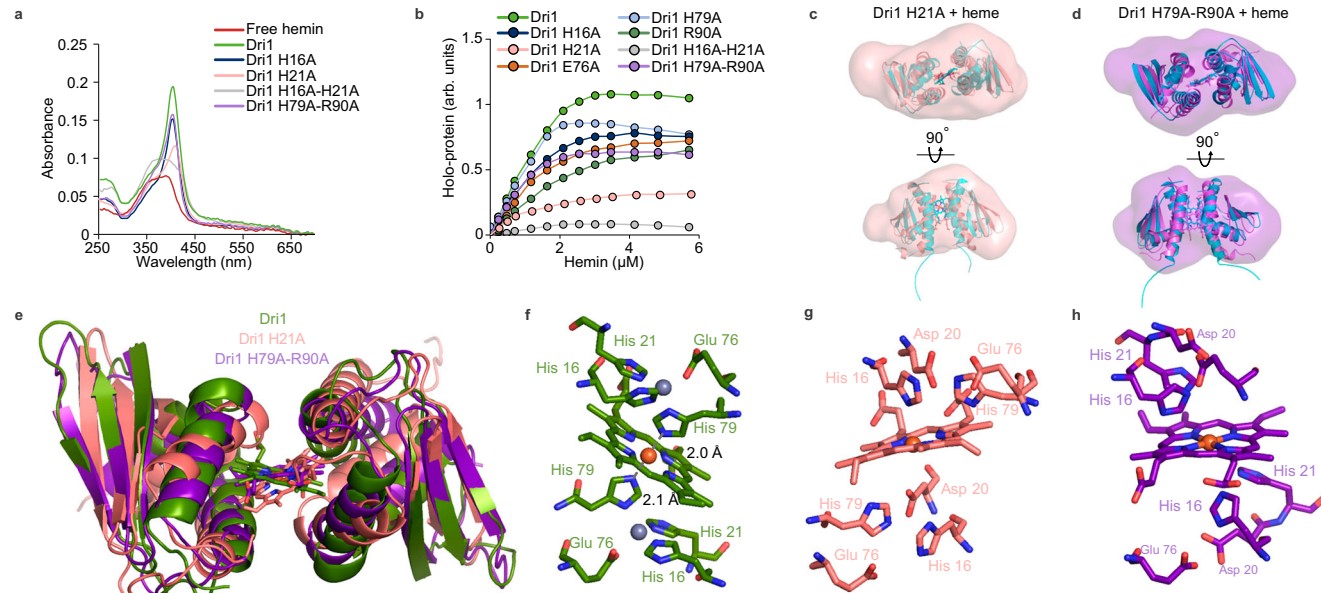

**Fig. 4 | Mutation of key histidine residues alters heme coordination. a** UV-Vis absorption spectra of Dri1 and amino acid variants with hemin. **b** Heme-binding curve of Dri1 and variants with increasing concentration of hemin. **c-d**, SAXS-MD best fit model superimposed onto the crystal structures (cyan) of Dri1 His21Ala (**c**) and Dri1 His79Ala-Arg90Ala (**d**) and overlaid onto DENSS envelopes calculated from SAXS to $q_{max} = 1.0$ Å⁻¹. Model to data fit for the SAXS-driven MD models and crystal structures are shown in Supplementary Fig. 9d and j, respectively. **e** Super-position of dimeric structures obtained from crystallography: Dri1 (green), Dri1 His21Ala (light pink), and Dri1 His79Ala-Arg90Ala (purple). Respective hemes at the dimer interface are shown in stick representation. **f–h** Selected side chains of Dri1 and variants involved in heme binding and interactions are shown (sticks) and labeled. Structures were determined from crystals. **f**, Heme iron is axially ligated by His79 which in turn is coordinated by a tetrahedrally bound zinc ion to form the WT Dri1+heme complex. **g** Substitution of His21 results in a shifted interaction and a rotated heme, preventing His79 from axially ligating the heme Fe. Unlike the 2 Å distance between His79 and Fe in Dri1, the nearest side chain to the heme iron in Dri1 His21Ala is Asp20 (~4 Å). **h** Dri1 His79Ala-Arg90Ala shows a similar organization to Dri1, although loss of the Arg90 side chain leads to 180° rotation of the heme and orients the propionate groups toward Arg12 near the N-termini. His21 and His16 residues weakly interact with heme Fe ~4 Å away. Source data are provided as a Source Data file.

from the two His79) results in a slightly better fit with 4 Fe-N at 2.01 Å and 2 Fe-N at 2.12 Å, respectively (Fig. 3g). While the resolution of EXAFS data does not allow high confidence in the distances in this split-coordination fit, it is consistent with the distribution of Fe-N(heme) and Fe-N(His79) distances observed in the crystal structure. The longer distance data are fit with Fe-C and Fe-N-C multiple scattering paths arising from the porphyrin ring (Supplementary Fig. 5b). One Fe-Zn path can be fit to the Zn-Dri1 +heme dataset, which does not fit the apo-Dri1 +heme data. This Fe-Zn path at 5.54 Å is in reasonable agreement with the crystal structure and is consistent with the existence of this zinc-mirror binding site (Fig. 3f–g, Supplementary Fig. 5a-c). Other possible scattering interactions that could give rise to this peak were tested (Supplementary Fig. 5a, d–e) and found to be incompatible on the basis of their phase, as observed in a comparison with the Fourier-back transform of the peak centered at ~5.3 Å in the EXAFS data. In the apo-Dri1 +heme, the data are best fit with 5 Fe-N paths at 2.02 Å, with 1 distant N/O observed at 2.51 Å likely from a weakly bound water molecule (Supplementary Fig. 5f). Additionally, the fitting of the EXAFS data using the WT proteins disregarding the Fe-Zn path resulted in a reduced fit quality (R factor increased by 0.04) (Fig. 3g and Supplementary Fig. 5c, g, h). The difference in first shell coordination directly indicates a role of zinc ions in bis-histidine heme binding.

Since Zn is not essential for heme binding (Supplementary Fig. 6), we compared apo-Dri1 with Zn-Dri1 and Co-Dri1 to better understand the role of zinc in heme binding. Zinc, but not cobalt, appears to significantly increase the heme-binding capacity and marginally increases the heme-binding affinity (Supplementary Fig. 6, Supplementary Fig. 7). The effect of the metal ions on the heme-iron environment was further characterized with X-band continuous-wave electron paramagnetic resonance spectroscopy (CW-EPR) measurements (Fig. 3h). Apo-Dri1 with heme exhibits a high-spin signal ($g = 6$;

$S = 5/2$). Zn-Dri1 with heme, however, has a significantly diminished $g = 6$ signal and exhibits features at $g = 2.75$, 2.27, and 1.72, corresponding to low-spin iron species ($S = 1/2$). The Co-Dri1 with heme has similar, albeit shifted, features at $g = 2.94$ and 2.27 (the high field signal was not observed). Such high- and low-spin signals are comparable to those of histidine-ligated hemoproteins in penta- and hexa-coordinated states, respectively[29–31]. The rhombic spectra of the Zn- and Co-Dri1 with heme are indicative of a conversion to a bis-axial His-heme ligation upon zinc or cobalt incorporation, with at least one of the histidine ligands being deprotonated[32].

## Mutation of key histidine residues alter heme coordination

Guided by the crystal structure, we next probed the role of key amino acids in heme binding (Fig. 4a, b, Supplementary Fig. 8). We observed that all variants with a single amino acid substitution retained heme binding to some extent. Both His21Ala and His21Ala-His79Ala variants displayed a red-shifted Soret peak at 415 nm, which corresponds to a reduced heme iron or a modified heme coordination (Figs. 2b, 4a, b, Supplementary Fig. 8). The His16Ala-His21Ala mutation completely abolished heme binding, similar to His9 and His14 mutations in ChuZ[21]. His79Ala and His79Ala-Arg90Ala surprisingly bound heme (Fig. 4a, b). To better comprehend the ability of His21Ala and His79Ala-Arg90Ala to bind heme, the two variants were subjected to SEC-SAXS, CW-EPR, and macromolecular crystallography. SEC-SAXS studies indicate little to no changes in the dimer structure (Fig. 3a–d). CW-EPR spectra of His79Ala-Arg90Ala show that the heme iron remains in the high-spin ($g = 6$; $S = 5/2$) state regardless of zinc addition and is not bis-axially coordinated by amino acid side chains (Fig. 3h).

Crystals of His21Ala and His79Ala-Arg90Ala, both heme bound, were successfully obtained and solved out to 2.85 Å and 2.90 Å, respectively, with similar folds as the wild-type (WT) structure and

conformations comparable to those obtained by SAXS-driven MD simulations (Fig. 4c, d, Supplementary Fig. 9–11, Supplementary Table 2). Despite the lower resolution and crystal disorder (high Wilson B factors), the data are sufficient for elucidating heme-binding interactions. His21Ala exhibits a shifted binding interface with a rotated heme ring (-34°) relative to WT Dri1 (Fig. 4e–g, Supplementary Fig. 10). Calculated Polder OMIT maps with the heme, His residues, and bulk solvent excluded (Supplementary Fig. 12a) in addition to higher ligand B-factors (Supplementary Table 1) reflect a loosely bound heme. The loss of His21, and presumably zinc binding, prevents axial ligation to the heme iron by His79. Instead, Asp20 seems most likely to be the side chain interacting with the heme iron, though the distance between the carboxyl functional group and iron is long (-4 Å Fe - O distance). This coordination distance can explain the increased heme rotation and mobility observed in MD simulations (Supplementary Fig. 10), resulting in a weaker bonding interaction and lower heme-binding ability (Fig. 4a, b). The His21Ala structure further suggests that zinc ions could bind via tetragonal ligation by His16, Asp20, Glu76, and His79 residues. However, since the Asp side chain can only have one binding interaction (as opposed to the two nitrogen sites in imidazole), zinc-binding might prevent Asp20 from interacting with the heme iron. Such a hypothesis is in line with the abrogation of heme binding by the addition of excess zinc to His21Ala (Supplementary Figs. 6, 7).

By comparison, the structure of His79Ala-Arg90Ala with heme has a similar dimer fold and interface as WT Dri1. Despite high R factors ($R_{work}/R_{free}$ = 0.311/0.326) due to translational non-crystallographic symmetry (Supplementary Table 1), similar structural conformations were also obtained by SAXS-driven MD simulations (Fig. 4d, Supplementary Figs. 9g-l, 10–11, Supplementary Table 2). A slight rotation of -30° of the plane of the heme is observed owing to the loss of His79 (Fig. 4e, f, h, Supplementary Fig. 10b). The remaining His side chains are equidistant from the heme iron (-4 Å N – Fe distances), likely contributing a weak interaction with the heme iron. Curiously, the absence of Arg90 permits the heme cofactor to rotate 180° such that the propionate groups interact with Arg12 at the N-termini (Supplementary Fig. 12b). The slight planar rotation of heme with respect to the monomers observed in MD simulations (Supplementary Fig. 10) can explain the relatively similar heme-binding ability of His79Ala-Arg90Ala to that of the WT protein (Fig. 4a, b). Additionally, we observed that the addition of zinc increases its heme-binding affinity and capacity (Supplementary Figs. 6, 7). Hence, the function of His79 can be compensated for by the other His residues; loss of the His79 axial ligands allows His16 and His21 to interact with the heme iron, therefore highlighting the variability in heme binding when His79 is absent. Since His16, His21, and His79 are absolutely conserved in all cyanobacterial Dri1 orthologs analyzed (Fig. 1c), the precise manner in which WT Dri1 binds heme may be critical to its biological function[31].

## Dri1 interacts with a subunit of the succinate dehydrogenase complex

Given the ability to bind heme but not degrade it, we hypothesized that Dri1 could either function as a heme chaperone or as a heme-dependent regulatory protein. For example, like GBP, which also binds heme and lacks heme-degradation activity, Dri1 could act as a heme-dependent negative feedback regulator of the tetrapyrrole biosynthetic pathway. To investigate this, we targeted enzymes catalyzing early reactions of the tetrapyrrole biosynthetic pathway in cyanobacteria by Y2H (Supplementary Fig. 13a). No interaction was detected between Dri1 and HemA (a glutamyl-tRNA reductase), HemB (an ALA dehydratase), or HemL (homolog of the plant GluTR1), suggesting a functional divergence between non-catalytic DRI-containing proteins from different phyla.

Next, we mined publicly available protein-protein interaction data to identify possible Dri1 interactors. Five protein candidates were found in a high-throughput Y2H study[24]: slr0665, sll1625, slr1020,

sll1946, and sll0785. Interestingly, most of the proteins identified, with the exception of slr1020, are iron-sulfur cluster binding proteins or iron-related proteins; slr0665 is an aconitate hydratase, sll0785 is annotated as 7,8-didemethyl-8-hydroxy-5-deazariboflavin synthase, ELP3 is an elongator acetyltransferase, and sll1625 (SdhB1) is an iron-sulfur cluster binding subunit of the succinate dehydrogenase complex. However, we were only able to reproduce the physical interaction between SdhB1 (sll1625) and Dri1 (Fig. 5a).

Given the specific interaction between Dri1 and SdhB1, we performed a phylogenomic analysis of the iron-sulfur cluster binding subunit SdhB (Supplementary Fig. 13c-g) and a co-occurrence analysis between DRI, SDH subunits, and PNPOx-like domain (found in GBP and HugZ) throughout the tree of life (Fig. 5b). Through the co-occurrence analysis, we determined that while homologs of the four main SDH subunits, as described for *E. coli*[33], yeast[34], human[35] and plants[36] (the cytoplasmic flavoprotein subunit SdhA, the cytoplasmic iron-sulfur cluster binding SdhB, and the heme-binding membrane-anchored subunits SdhC, and SdhD), generally exist in eukaryotes and bacteria, only two homologous subunits are found in cyanobacteria (SdhA and SdhB). HdrB (also referred to as SdhE[37]) appears to functionally substitute for SdhC and SdhD as characterized in *Campylobacter jejuni* and *Synechocystis*[7,37,38] (Fig. 5b, c). In addition to the substitution of SdhC/SdhD by HdrB, there are two SdhB homologs present in *Synechocystis*; sll1625 (SdhB1), which possesses the C-terminal extension conserved in all cyanobacteria, and sll0823 (henceforth denoted as SdhB2) (Fig. 5b, Supplementary Fig. 13b-d). Furthermore, co-occurrence analysis highlights the presence of SdhB1 and Dri1 orthologs strictly in cyanobacteria (Fig. 5b).

Interactions between Dri1 and other cyanobacterial SDH subunits were screened by Y2H. Positive interactions were only found with SdhB1 and, to a lesser extent, SdhB2 (Fig. 5a). We also tested the presence of this interaction in vivo (Fig. 5d). In *Synechocystis*, Dri1 and SdhB1 were expressed with C-terminal-Strep-tag II and -FLAG tags, respectively. SdhB1-FLAG was detected only in the presence of Dri1. This result indicates that SdhB1 coeluted with Dri1, supporting the formation of a complex in vivo. The addition of heme significantly decreased the amount of co-eluting SdhB1-FLAG (Fig. 5d). Based on the structure of Dri1, we propose that heme binding leads to a Dri1 homodimer that can no longer bind to SdhB1.

We used homology modeling and docking analyses to predict candidate residues for a heme-free Dri1-SdhB1 heterodimer (Fig. 5e–g). SdhB1 residues, namely Cys214, Tyr215, and Glu219, were computationally predicted to interact with Dri1 zinc-binding residues (His16, His21, His79, and Glu76) (Fig. 5g). In the Y2H assays, mutation of His79Ala and Arg90Ala, both single and simultaneous replacements, did not hinder Dri1-SdhB1 dimer formation. However, mutations of the metal-binding site residues (His16Ala, His21Ala, or Glu76Ala) led to loss of an interaction (Fig. 5a), suggesting their involvement in complex formation. Additionally, the mutation of either Cys214 or the simultaneous mutation of Tyr215 and Glu219 residues from SdhB1 completely prevented the interaction between the two partners (Fig. 5e-h), supporting the computationally based heme-free Dri1-SdhB1 dimer model.

## Dri1 is directly or indirectly involved in photosynthetic and respiratory electron flow

Given the interaction of Dri1 with SdhB1, we hypothesized that Dri1 could play a role in respiration and photosynthesis. We constructed mutant strains in *Synechocystis* by leveraging the marker-less genome editing CRISPR-Cas12a[39,40]. The deletion of *dri1* (Δ*dri1*) had no effect on growth under photo-autotrophic conditions compared to the WT (Fig. 6i). We compared the chlorophyll fluorescence of these two strains as a proxy for any disruption or modification of components in the electron transfer chains. In cyanobacteria, the redox state of the plastoquinone (PQ) pool is in equilibrium with $Q_A$ (the first electron-

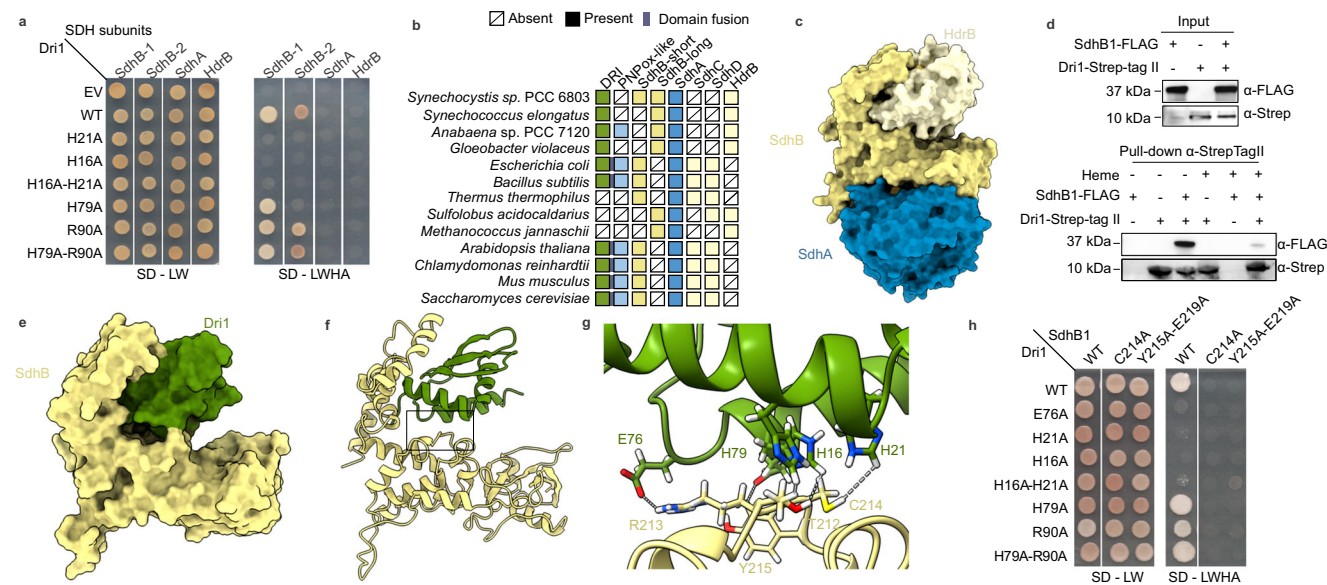

**Fig. 5 | Dri1 interacts with a subunit of the succinate dehydrogenase complex.**
**a** Y2H interaction assay of Dri1 and amino acid variants ('prey' fused to GAL4 activation domain, pGADT7-AD) with SDH subunits ('bait' fused to DNA-binding domain, pGBKT7-BD). **b** Co-occurrence analysis throughout different phyla of DRI domain, split β-barrel domain (PNPox-like), and SDH subunits. Proteins were predicted from whole genome sequencing databases using *Synechocystis* proteins as the query. For SDH subunits SdhC and SdhD, sequences from *Escherichia coli* were used as query (absent in cyanobacteria). Colors correlates with the protein colors used in other Figs. **c** AlphaFold complex model (surface representation) of the SDH complex from *Synechocystis* composed of three subunits SdhA (blue), SdhB1 (yellow), and HdrB (white). **d** Strep-tag II protein pull-down assay in *Synechocystis* expressing SdhB1-FLAG or Dri1-Strep-tag II C-terminal fusions in the presence or absence of heme. Protein coelution was analyzed by immunoblot using antibodies targeting either FLAG or Strep-tag II, as mentioned. The input panel represents the protein presence before incubation with Strep-Tactin Sepharose resin used for the co-elution pull down. Representative immunoblots of three independent experiments is shown. **e–f** Surface representation of the computationally predicted structure of a Dri1-SdhB1 heterodimer using molecular docking simulation in the absence of heme. **g** Side chains (sticks representation and labeled) predicted to take part in the interaction between Dri1 (green) and SdhB1 (yellow). **h** Y2H interaction assay between Dri1 variants ('prey', pGADT7-AD) and SdhB1 variants ('bait', pGBKT7-BD). Y2H interactions images displayed are representative of 3 independent clones. Source data are provided as a Source Data file.

accepting PQ in PSII). Therefore, since chlorophyll fluorescence reflects the redox state of $Q_A$, it also indirectly reflects the redox state of the PQ pool[41]. Comparison of chlorophyll fluorescence kinetics revealed an increased minimum ($F_0'$) and maximum ($F_{m'dark}$) chlorophyll fluorescence in the dark-acclimated state, as well as a higher steady-state fluorescence under actinic light ($F_s$) in $\Delta dri1$ compared to the WT (Fig. 6a). An apparent chlorophyll fluorescence increase could suggest an altered pigment composition, a possible impairment of PSII function, or a higher level of $Q_A^-$ (i.e., a more reduced PQ pool)[42–44]. To account for any putative influence of differences in the pigment composition (e.g., highly fluorescent phycobilisomes) of the strains, we measured their absorbance spectra normalized to the optical density at 750 nm (Fig. 6b). No difference was observed between the two strains, suggesting that the higher fluorescence is not due to a modified pigment composition. This modified chlorophyll fluorescence was not the result of an altered accumulation of major photosynthetic membrane complexes (PSI and PSII) triggered by the absence of Dri1, as verified qualitatively by blue native PAGE (BN-PAGE) (Fig. 6c). Calculation of the $Q_A$ redox state (1-qP) at the end of the 5 min of actinic light (Fig. 6a) showed a higher value (0.81) in the $\Delta dri1$ mutant than in the WT (0.71). Therefore, the increased chlorophyll fluorescence displayed by $\Delta dri1$ implies a modified PQ redox state compared to the parental strain.

The SDH complex catalyzes the oxidation of succinate into fumarate as part of the TCA cycle[45]. It is one of the main substrate-dependent electron donors in cyanobacteria, alongside NDHs, by providing electrons from succinate to reduce the PQ pool[8]. We compared the PQ pool redox states between the WT parental strain, $\Delta dri1$, and a strain where the codon for His21 was mutated to code for alanine (referred to as *dri1*(H21A)) (Fig. 6d). His21 is one of the residues involved in the Dri1-SdhB1 interaction. As described above, the

measured fluorescence rise is dependent on the accumulation of $Q_A^-$. By blocking oxidases with potassium cyanide (KCN) in the dark, electrons from the reduced PQ pool are back transferred to $Q_A$. Additionally, because of the absence of actinic light, PSII does not contribute electrons to the reduction of $Q_A$. Therefore, the rise of fluorescence in the presence of KCN in darkness is an indirect assessment of the PQ pool reduction kinetics by respiratory complexes[8,9]. Our results show a faster accumulation of $Q_A^-$ for both $\Delta dri1$ ($18.54 \pm 1.95$ s) and *dri1*(H21A) ($72.87 \pm 25.68$ s) compared to the WT parental strain ($122.78 \pm 1.70$ s) (Fig. 6d). By comparison, the SDH double null mutant $\Delta sdhB1$ $\Delta sdhB2$ (deletion of sll1625 and sll0823) leads to a slower reduction of the PQ pool[8,9]. Given the interaction between Dri1 and SdhB1, Dri1 may be a negative regulator of SDH activity. Consistent with this hypothesis, SDH activity, assessed by succinate-dependent 2,6-dichlorophenolindophenol (DCPIP) reduction activity in membranes[9,38] isolated from strains grown with or without glucose, is higher in $\Delta dri1$ (Fig. 6e-f).

## Dri1 may be involved in multiple cellular processes
While no differences in growth were observed in the presence or absence of glucose, a phenotype was apparent in the presence of the herbicide 3-(3,4-dichlorophenyl)−1,1-dimethylurea (DCMU) that inhibits electron transfer from PSII to the PQ pool (Fig. 6g). The WT strain grew poorly in the presence of DCMU, but deletion of *dri1* rescued growth (Fig. 6g,j,k). We also tested the impact of amino acid mutations on this phenotype. Concordant with the in vitro heme-binding results and interaction with SdhB1, His21Ala and His16Ala-His21Ala mutants phenocopied the $\Delta dri1$ strain (Fig. 6h−k). To test if this phenotype is due to the function of Dri1 as an inhibitor of SDH, we also analyzed growth of the $\Delta sdhB1$, $\Delta sdhB2$, and $\Delta sdhB1$ $\Delta sdhB2$ mutants. However, the SdhB mutants but not mutants of NDH ($\Delta sll1484$, $\Delta sll0223$, and

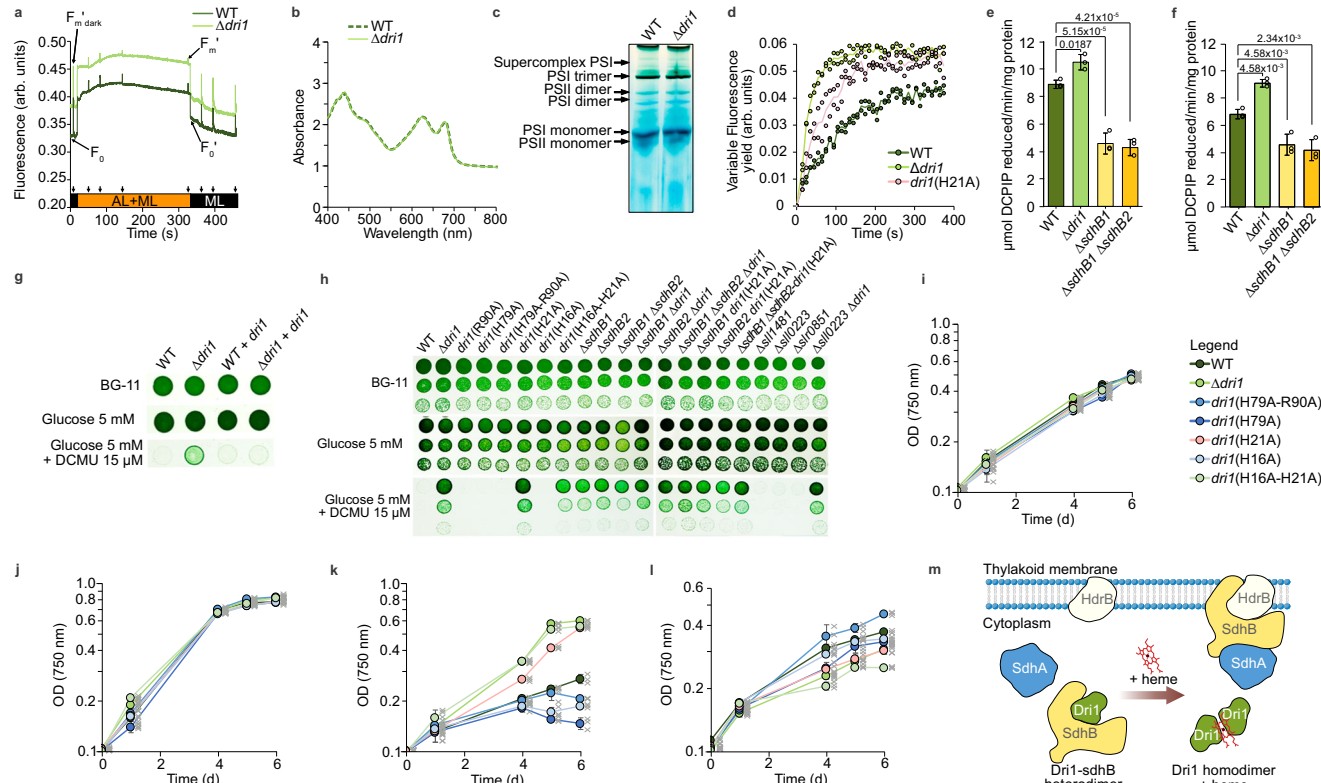

**Fig. 6 | Dri1 is involved directly or indirectly in photosynthetic and respiratory electron flow. a** Chlorophyll *a* fluorescence as measured with a pulse-amplitude modulated fluorometer in WT *Synechocystis* and Δ*dri1* mutant. Cells were dark-acclimated for 30 min prior turning on the non-actinic measuring light (ML) followed by turning on orange actinic light (AL, 620 nm) for 5 min, and 2 min of dark relaxation period. $F_0$ - minimum fluorescence in the dark. $F_{m'dark}$ - maximum fluorescence in the dark. $F_{m'}$ - maximum fluorescence in the light. Arrows indicate saturating light pulses. **b** Whole cell absorption spectra of WT (dark green dashed line) and Δ*dri1* (light green plain line), normalized to OD 750 nm. **c** BN-PAGE of membrane-bound protein complexes of WT and Δ*dri1* cells grown in BG-11 for 6 days. Gel image is representative of two independent experiments. **d** Plastoquinone pool reduction kinetics over 6 min of WT, Δ*dri1*, and *dri*(H21A) in the dark to block PSI and PSII activity and in the presence of KCN to block the activity of oxidases (*n* = 2 biological replicates, individual datapoints are shown,

lines represent the means). **e–f** SDH activity using DCPIP reduction of isolated membranes from *Synechocystis* grown without (**e**) or with (**f**) glucose for 6 days (mean ± SD, *n* = 3 biological replicates, *p*-values calculated using a two-sided pairwise t test with holm adjustment). **g** WT, Δ*dri1*, and complemented Δ*dri1* expressing *dri1* in trans grown on agar-solidified BG–11 with or without glucose and DCMU. The image is a composite of three separate plates incubated together and imaged at the same time. **h** Cultures of three ten-fold dilutions of WT and mutated strains grown as in (**g**). Images displayed are representative of several independent experiments. **i–l** Growth curves of cultures grown in BG–11 (**i**) BG–11 + glucose (**j**), and BG–11 + glucose + DCMU (**k**) for 6 days at 30 °C. (**l**) Growth curve in iron-deficient BG–11 in the presence of the iron chelator DFOA (mean ± SD, individual datapoints are depicted by crosses, *n* = 3 biological replicates). **m** Dri1 is a heme-dependent post-translational regulator of SdhB. Source data are provided as a Source Data file.

Δ*sll0851*), also showed increased tolerance to DCMU (Fig. 6h), suggesting that the DCMU-insensitivity of the Δ*dri1* may be separate from its role as an inhibitor of SDH. Since Dri1 is a heme-binding protein and could function to sense heme levels, we also tested the growth of Δ*dri1* during iron limitation. Deletion of *dri1* or mutation of His16Ala-His21Ala led to a decreased growth of cells when exposed to the iron chelator deferoxamine (DFOA; Fig. 6i,l).

## Discussion

Our identification and characterization of Dri1, containing a domain formerly of unknown function, has revealed a remarkable heme-zinc site along with an intriguing heme-dependent regulatory function. Structural studies of the Dri1-heme homodimer illustrate a heme-binding interface driven by two zinc ions ligated by histidine side chains (Fig. 2g-h). In turn, two zinc-bound histidine residues, one from each monomeric chain, coordinate to the iron center of a single heme cofactor (Fig. 2h). These zinc ions may help catalyze the formation of the imidazole-iron bonds by increasing the basicity of the histidine Nε2 atoms in a charge-relay fashion and/or play a structural role responsible for the precise positioning of heme between the two monomers. Indeed, mutation of the coordinating amino acids or absence of zinc impact how heme is bound (Fig. 4), suggesting that this site evolved to

bind heme in a precise way. The unusual zinc-mirror heme coordination of Dri1 is reminiscent of, but distinct from, some previously characterized structures. For instance, a doubly deprotonated histidine side chain also occurs in Cu-Zn superoxide dismutase, where the imidazole bridges copper and zinc ions[46]. In the WWD domain of the CcsBA heme exporter[47], the heme iron is protected by the presence of a histidine residue on each side of the porphyrin plane. However, we report here a protein possessing a two-fold symmetric zinc-histidine-heme iron-histidine-zinc coordination center at the interface of a homodimer.

Carbon and iron cycles are intertwined as iron is an essential cofactor for photosynthesis and respiration. The impact of this relationship is obvious in the oceans, where iron is one of the most limiting nutrients in primary productivity[48,49]. Therefore, it is not surprising that iron-sensing post-translational regulatory proteins have evolved to fine-tune essential, energetic, and iron-expensive processes. Based on our results, we describe a model of the signaling function of Dri1, linking heme homeostasis and carbon metabolism[11] (Fig. 6m). We hereby propose that Dri1 is a heme-dependent post-translational regulator of SdhB, a subunit of SDH. Given the high affinity of Dri1 for heme, Dri1 could act as a direct heme sensor. In conditions of iron deficiency, cellular levels of iron cofactors (e.g., heme and iron-sulfur

clusters) become limiting. Therefore, under low cellular heme availability, Dri1 binds to SdhB, preventing SdhB docking to the membrane-bound domain HdrB (Fig. 6m). Subsequently, the formation of a fully functional SDH complex is inhibited, reducing the contribution of SDH to the TCA cycle and to the photosynthetic (linear and cyclic electron flow) and respiratory electron transfer chains. Under iron-replete conditions, heme synthesis and abundance increase, which triggers heme-dependent Dri1 dimer formation. As a result, SdhB is free to bind HdrB and SdhA and form a functional SDH membrane complex. Such a hypothesis is reminiscent of the function of DRI-containing GBP, where the interaction between GBP and GluTR1 is inhibited by heme. However, while in plants heme binding to GBP triggers the degradation of its target[15], in cyanobacteria, heme lifts the negative regulation of SdhB by Dri1. The in vitro and in vivo interaction between Dri1 and SdhB, the effect of heme on this interaction, and the impact of the Δ*dri1* mutation on PQ pool reduction and SDH activity are all consistent with this model.

We also observed that in the presence of glucose and the PSII inhibitor DCMU (inducing photo-heterotrophic conditions) the WT strain grew poorly, but deletion of *dri1* enables photo-heterotrophic growth. These phenotypes suggest that the WT strain is dependent on PSII for growth, but the *dri1* mutant is not. As a heme-dependent repressor of SDH activity, the deletion of *dri1* and subsequent increase in SDH activity, the TCA cycle, and cyclic electron flow could explain why growth with glucose is no longer dependent on PSII. Surprisingly, however, we observed that the deletion of either *sdhB1* or *sdhB2* also enabled growth on glucose when PSII was inhibited. One hypothesis is that the photo-heterotrophic phenotype of the Δ*dri1* strain is independent from its function as a repressor of SDH activity. Another possibility is that the SdhB-Dri1 heterodimer could have an additional, unidentified function in Fe-dependent regulation of glucose metabolism. Other members of the TCA cycle have iron-dependent bifunctionality[50]. In Metazoa, aconitase catalyzes the isomerization of citrate to isocitrate, while during iron starvation, it acts as an iron-regulatory protein. In this role, aconitase binds to messenger RNA stem loops called iron-responsive elements (IRE) to prevent iron storage[50]. Another hypothesis is that the ability of the Δ*sdhB1* and Δ*sdhB2* strains to grow in the presence of DCMU is mediated by a separate compensatory mechanism compared to the Δ*dri1* strain, suggesting that different mutations can lead to a switching in the ability of *Synechocystis*, and likely other cyanobacteria, to grow with glucose as the sole source of carbon.

In conclusion, we provide evidence for the role of Dri1 as a heme-dependent post-translational regulator of SDH and describe a distinct zinc-mirror heme site, ultimately linking heme sensing to the reduction of the PQ pool. Given the absolute conservation of this protein across cyanobacteria, but not in plants or algae, the evolution of a physical separation afforded by specialized organelles for photosynthesis and respiration (the chloroplast and mitochondrion, respectively) presumably made Dri1 function unnecessary. However, homologous proteins containing DRI that are involved in heme homeostasis, either as a domain in heme-degradation proteins or in regulatory proteins, are prevalent throughout different phyla, including plants, suggesting a crucial and conserved role of this domain in heme homeostasis.

## Methods
### Phylogenomics analyses
The protein sequence encoded by *dri1* (*ssr1698*) from *Synechocystis* (UniProt: P73129) was used to search the UniProt database[51] using hmmsearch (Database: uniprotrefprot, version 2021_04, downloaded on 2022-01-15)[52]. The resulting hits were then mapped to UniProt's UniRef90 clusters[53], and only those sequences that are UniRef90 cluster representatives were used to build a phylogenetic tree. A multiple sequence alignment file for Dri1 (encoded by ssr1698) and

homologs was built with hmmer[52] and used to construct a maximum likelihood tree with IQ-TREE[54]. The phylogenetic tree was visualized and annotated with iTOL[55]. Branches with less than 50% bootstrap support were deleted. Sequence information, alignment file, and tree file can be found in Supplementary Data 1. WebLogo[56] was used to build sequence logos of DRI from each distinct protein family clade. EFI-GNT[57] was used to collect gene neighbors. Domain boundaries were determined with InterProScan[58].

### Plasmids and strain construction
Cyanobacterial strains used in this study are listed in Supplementary Data 2 and are derived from *Synechocystis* sp. PCC 6803 Glucose-Tolerant (original strain was provided by Dario Leister and originated from H. Pakrasi (Washington University, St. Louis)). Knock-out mutations for *dri1* (*ssr1698*), *sdhB1* (*sll1625*), *sdhB2* (*sllO823*), *sll1481*, *sllO223*, and *slr0851*, and site-directed mutations were generated using CRISPR-Cas12a developed for cyanobacteria[39,40]. Primers and plasmids generated are listed in Supplementary Data 2. For CRISPR-Cas12a mutagenesis, 20 nucleotides long sgRNAs were designed using the gRNA-SeqRET tool (https://grna.jgi.doe.gov/grna.html)[59] and synthesized as complementary single-stranded oligonucleotides. The sgRNAs were annealed and ligated into AarI-digested pCpf1b vector. For gene deletion, repair templates consisted of two 750 bp homology arms upstream and downstream of the start and stop codons of the target, respectively. To introduce point mutations, 750 bp fragments from upstream and downstream of the mutagenesis site were amplified, and the codon of the amino acid to be mutated was modified accordingly by mutagenic PCR. Additionally, the PAM sequence was also mutated with a synonymous codon to prevent Cas12a cleavage of the vector. Repair templates were PCR amplified and cloned by Gibson assembly into the sgRNA-containing pCpf1b plasmid digested by BamHI and BglII.

To generate a plasmid for overexpression in *Synechocystis*, pCpf1b was digested with BamHI and SapI to remove sequences encoding for Cas12a. The promoter and terminators of *psbA* from *Synechocystis* were PCR amplified using primers NG0052 / NG0053 and NG0054 / NG0055 from genomic DNA (Supplementary Data 2). Purified promoter and terminator PCR fragments were cloned into BamHI-SapI-digested pCpf1b by Gibson assembly, generating the overexpression vector pNG055. CDS sequence of *dri1* was PCR amplified using primers NG0056 / NG0057 from gDNA and cloned into the EcoRI-digested pNG055 plasmid by Gibson assembly to create pNG056. *dri1*-eYFP fusion was generated by amplifying *dri1* from gDNA using primers NG0056 / NG0022 and amplifying eYFP from pGWB442 using primers NG0021 / NG0058. The *dri1* and eYFP genes were assembled and cloned in pNG055 by Gibson assembly to generate pNG057. To generate overexpression plasmids for *dri1*-Strep-tag II, *dri1* was amplified from gDNA using primers NG0056 / NG0311 to add a C-terminal Strep-tag II and cloned into EcoRI-digested pNG055 by Gibson assembly to generate pNG058. To create *sdhB1*-3×FLAG fusions, *sdhB1* was amplified from gDNA using primers NG0313 / NG0314 and assembled by Gibson assembly with ultramers NG0304 / NG0314 to be cloned into pNG055 to generate pNG059.

For purification of Dri1-His tag fusion proteins, the CDS of *dri1* was codon optimized for *Escherichia coli*, synthesized by Twist Bioscience (South San Francisco, CA), and cloned by Gibson assembly into the NdeI restriction site of pET11e vector carrying a C-terminal TEV cleavage site followed by 10 × His-Tag sequence. Site-directed mutagenesis was achieved using the megaprimers method using the primers listed in Supplementary Data 2, and mutated *dri1* sequences were cloned into NdeI-digested pET11e by Gibson assembly.

### Protein purification
Protein overexpression was carried out in *E. coli* BL21 (DE3) cells by autoinduction[60] at 17 °C for 18 hrs. Pellets from 0.5 L cultures were

collected by centrifugation at 2500 x $g$ for 20 min the following day and either frozen or immediately lysed for purification. Pellets were lysed either chemically or by sonication – both methods resulted in high yields of pure protein. Chemical lysis was performed using a lysis buffer (75% B-PER (Thermo Scientific), 25% Y-PER (Thermo Scientific), 50 mM HEPES, pH 7.5, 200 mM NaCl, 1 mg/mL lysozyme, 10 mg mL$^{-1}$ benzonase nuclease, and cOmplete protease inhibitor cocktail (Roche)) and incubated on ice for 30 min. For cell lysis by sonication, pellets were resuspended in a buffer of 50 mM HEPES, pH 7.5, 200 mM NaCl, 1 mg/mL lysozyme, and cOmplete protease inhibitor cocktail (Roche). Cells were lysed with 2–3 cycles of sonication (2 sec on, 2 sec off; 4 min in total per cycle, power level of 20–30%) on ice. Cell lysates were centrifuged at 40,000 x $g$ for 1 h at 4 °C to remove insoluble material. The supernatant was then loaded onto a column of Ni-NTA resin and allowed to bind for 20 min. The column was washed with 25 mM imidazole to remove nonspecific binding, and proteins eluted with 300 mM imidazole. Eluted proteins were treated with TEV protease to remove the purification tag and dialyzed into 50 mM HEPES, pH 7.5, 200 mM NaCl, 1 mM TCEP overnight at 4 °C. Cleaved His$_{10}$ tags were removed by reverse Ni-NTA. Owing to the metal-binding site in the protein, the column had to be washed with up to 100 mM imidazole to recover most of the tag-free protein. Proteins were then purified by size-exclusion chromatography (HiLoad 16/600 Superdex 200, Cytiva) in 50 mM HEPES, pH 7.5, 200 mM NaCl. Pure fractions were combined, concentrated to ~1 mM, aliquoted, and stored at -80 °C.

Samples for SAXS, CW-EPR, and XAS were prepared similarly. Briefly, proteins were either used as purified from size-exclusion chromatography, or proteins were first demetallated overnight with 15 mM EDTA. Treated samples were then exchanged into CHELEX-treated, metal-free buffer to remove EDTA. The absence of metal was verified by ICP-MS. For metal reconstituted samples, demetallated proteins were treated with 1 molar equivalents of ZnSO$_4$ or CoCl$_2$ dissolved in ultrapure water. For heme-bound samples, proteins were treated with 0.6–0.7 molar equivalents of freshly prepared hemin in 0.1 M NaOH. After at least 30 minutes, the mixture was neutralized with HCl and centrifuged to remove insoluble hemin. All samples were then buffer exchanged using an Amicon 10-kDa centrifugal filter (Millipore) or a PD SpinTrap G-25 column (Cytiva) to remove excess cofactors. Due to heme absorption at 280 nm, protein concentrations were determined by Bradford assay (Bio-Rad) or with the Qubit Protein Assay kit (Invitrogen). For SAXS, CW-EPR, and XAS, sample concentrations were ~1–1.5 mM.

## Inductively coupled plasma mass spectrometry (ICP-MS)

Protein samples were denatured with 69% nitric acid overnight at room temperature and diluted the following day with ultrapure water to a final concentration of ~5 μM in 2% nitric acid. Metal contents were determined by ICP-MS on a NexION 350D (PerkinElmer) calibrated with an environmental standard mix (N9307805, PerkinElmer), instrumental metal calibration standard (N9301721, PerkinElmer), and $^{89}$Y and $^{115}$In as internal standards (M1-ISMS-25, Elemental Scientific). Isotope abundances were measured using a Helium collision mode with kinetic energy discrimination. Protein samples (5 μM in 2% nitric acid) were prepared in triplicate as described above and injected by autosampler (500 μL). Metal occupancy percentages were determined from dividing the averaged readings by the corresponding protein sample concentrations.

## Heme-binding assays analysis and $K_d$ determination

Heme titration experiments were conducted similarly to a previously detailed protocol by Leung et al.[25] All spectra were collected in quartz cuvettes ($b = 1$ cm) on a NanoDrop One$^C$ Spectrophotometer (Thermo Scientific). Fresh hemin stocks (~0.2 mM to 0.6 mM) were prepared in 0.1 M NaOH in ultrapure water right before titration. A spectrum of ~3 μM hemin was then acquired before starting experiments to confirm

the concentration of the stock ($\varepsilon_{385\,nm} = 58.4\ mM^{-1}\ cm^{-1}$). Solutions of 10 mM ZnSO$_4$ and CoCl$_2$ were prepared in ultrapure water. Each titration experiment began with 5 μM of protein in 850 μL of buffer (50 mM HEPES, 200 mM NaCl). Unless otherwise specified, the buffer pH was 7.5. To prepare zinc/cobalt-treated protein, 50 μM of metals were added to de-metalated protein in CHELEX-treated metal-free buffer and incubated at room temperature for 8 minutes before adding hemin.

Heme titrations were done in 13 increments. The sample was mixed gently by pipette with every addition, and the spectra were collected after 3 minutes to allow for heme binding to reach a steady state. Final hemin concentrations were 2.5–10 μM depending on the binding affinity. In general, the endpoint was determined by the loss of spectral features, specifically the Soret peak (A$_{405–418\,nm}$) and shoulder (A$_{360\,nm}$), and by the overall spectra resembling that of unbound heme. Oxidized- and reduced-heme bound Dri1 spectrum were obtained by addition of excess ammonium persulfate and sodium dithionite, respectively. To determine whether Dri1 was able to degrade heme, UV-Vis absorption spectra kinetics were performed with 10 μM Dri1 and 10 μM hemin in the presence of 10 mM freshly prepared ascorbate as electron donor. Kinetics were started after addition of ascorbate, and monitored every 5 min for 140 min.

Heme-binding affinities were determined using a multivariate-curve resolution-alternating least squares (MCR-ALS) algorithm (MCR-ALS 2.0 toolbox)[61] in a MATLAB environment (R2019a; MATLAB, MathWorks, USA) followed by nonlinear curve-fitting to theoretical binding curves.

Owing to spectral feature overlaps, deconvolutions were necessary to obtain the pure spectra of each component and their weighted concentration profiles. For a given set of repeated titrations, the pure spectra should be common to all experiments, so simultaneous deconvolutions were implemented to help constrain the fits and calculate realistic spectra. This multiset data analysis was done by generating a column-wise augmented data matrix comprised of submatrices $\mathbf{D}$ ($m \times n$). Each submatrix $\mathbf{D}$ had $m$ rows of experimental spectra in the set, with the last row being the spectrum of unbound heme at ~3 μM (for a total of 15 spectra), and $n$ columns of absorption values (i.e., absorbance measurements over the wavelength range of 245–750 nm). The augmented matrix was then deconvoluted into a product of concentration profiles (submatrices $\mathbf{C}$) and a single, common set of pure spectra (matrix $\mathbf{S}$) with additional submatrices $\mathbf{E}$ of the residuals:

$$\begin{bmatrix} \mathbf{D_1} \\ \vdots \\ \mathbf{D_n} \end{bmatrix} = \begin{bmatrix} \mathbf{C_1} \\ \vdots \\ \mathbf{C_n} \end{bmatrix} \mathbf{S^T} + \begin{bmatrix} \mathbf{E_1} \\ \vdots \\ \mathbf{E_n} \end{bmatrix} \qquad (1)$$

Non-negativity constraints (fnnls; fast non-negative least squares algorithm) were applied to all submatrices. Equal spectra intensity normalizations were chosen to avoid scale indeterminacies during ALS optimization, and the convergence criterion was set to 0.05. All data were fit with 4–5 components to obtain pure spectra of unbound heme, holoprotein, and apoprotein. Additional features were determined to be background components from precipitated protein and unknown side processes. Moreover, hemin additions and measurements (conducted after maximum binding was attained) were discarded to avoid any fitting complications from these unknown side reactions (range of fits indicated in Supplementary Fig. 6c). In the cases where the extra components clearly resembled the holoprotein, the corresponding concentration profiles were summed together ("holo combined"; Supplementary Fig. 6); summation of similar components was done instead of reducing the number of components in the fit due to issues of calculating pure spectra in the latter. The resultant concentration profiles of free hemin and holoprotein were then fitted to

theoretical binding curves. Theoretical binding curves were derived as by Leung et al.[25] from the equation for $K_d$, where [apo], [holo], and [heme] are the molar concentrations of protein, protein with bound heme, and unbound heme, and $i$ is the index for each hemin addition:

$$K_d = \frac{[apo]_i \times [heme]_i}{[holo]_i} \quad (2)$$

This gave the following set of equations describing the relationship between protein and heme concentrations throughout the titration experiment, where $V_0$ is the initial volume of the buffer and protein, $V_i$ is the total volume of added hemin at increment number $i$, $p_0$ is the initial concentration of protein, and $h_0$ is the initial concentration of the hemin solution:

$$[apo]_i + [holo]_i = p_0 \times \frac{V_0}{V_0 + V_i} \quad (3)$$

$$[holo]_i + [heme]_i = h_0 \times \frac{V_i}{V_0 + V_i} \quad (4)$$

The explicit equations for the concentrations of protein with heme bound ([holo]) and unbound hemin ([heme]) were solved by using Eqs. (2), (3), and (4), yielding the following:

$$[holo]_i = \frac{1}{2}\left[\frac{p_0 V_0 + h_0 v_i}{V_0 + v_i} + K_d - \sqrt{\left(\frac{p_0 V_0 + h_0 v_i}{V_0 + v_i} + K_d\right)^2 - 4\frac{p_0 V_0 \times h_0 v_i}{(V_0 + v_i)^2}}\right] \quad (5)$$

$$[heme]_i = \frac{1}{2}\left[\frac{h_0 v_i - p_0 V_0}{V_0 + v_i} - K_d + \sqrt{\left(\frac{p_0 V_0 + h_0 v_i}{V_0 + v_i} + K_d\right)^2 - 4\frac{p_0 V_0 \times h_0 v_i}{(V_0 + v_i)^2}}\right] \quad (6)$$

Fits to the concentration profiles were optimized using arbitrary scaling factors $\alpha$ and $\beta$ to convert the concentrations into molar concentrations (i.e., $\alpha[holo]_i$ and $\beta[heme]_i$). As $h_0$, $V_0$, and $v_i$ were known a priori, the only unknown variables were $p_0$, $K_d$, $\alpha$, and $\beta$. (Generally, protein concentration $p_0$ cannot be accurately measured.) Using the function lsqcurvefit in MATLAB, these four coefficients were optimized to best fit the system of equations to the concentration profiles obtained from MCR-ALS.

## Crystallization and structure determination
Native, as-purified Dri1 crystals were grown at 25 °C by the sitting drop vapor diffusion method, using a 1:1 ratio of protein: reservoir solution containing 25% PEG 3350 and 100 mM bis-tris pH 5.5. The crystals were transferred to a reservoir solution containing 20% ethylene glycol before being flash-frozen in liquid nitrogen. The X-ray diffraction data were collected at the FMX (17-ID-2) beamline of NSLS-II (National Synchrotron Light Source II; Brookhaven National Laboratory, Upton, NY). Crystals of as-purified Dri1 diffracted to 2.35 Å resolution, and the data were processed with XDS-based FastDP[62]. The Matthews coefficient (VM) was calculated as 2.02 Å³ Da⁻¹, which corresponds to four monomers per asymmetric unit with an estimated solvent content of 45%. The structure of as-purified Dri1 was determined by molecular replacement with Phaser in the CCP4 program suite[63,64]. The C-terminal domain (residues Pro179 to Ser266) of the heme-binding protein from *Mycobacterium smegmatis* (PDB 5BNC) was used as a search model. The structure was refined by several rounds of iterative model building with Coot[65] and refinement in CCP4 module, REFMAC[66] (Supplementary Table 1).

The Dri1+heme complex was prepared by mixing heme to Dri1 at a molar ratio of 5:1 and crystallized with the same conditions as

as-purified Dri1. A 10 mM hemin stock solution was prepared in DMSO before mixing with Dri1. The structure of the complex was determined by rigid-body refinement of the as-purified Dri1 model followed by restrained refinement in REFMAC[66]. The difference Fourier map clearly showed interpretable electron density for the heme at the interface of two Dri1 subunits (Supplementary Fig. 3), and heme was modeled and refined accordingly (Supplementary Table 1). The presence of mild twinning (twinning fraction of 0.23 for h, -h-k, -l) contributed to the slight increase in the difference between Rwork and Rfree values (Supplementary Table 1). Additionally, disordered regions containing residues 6-9 and 67-72 exhibit poor electron density.

Initial crystallization hits of Dri1-His21Ala +heme were identified using the Hauptman-Woodward Institute standard screen[67]. Crystals were optimally grown by hanging drop vapor diffusion with well solutions of 0.1 M sodium citrate, 0.1 M potassium phosphate, pH 5, 34% PEG 8000. Crystals formed in a few days and were grown for a week, after which, they were cryoprotected briefly in well solution with 25% ethylene glycol before flash freezing in nitrogen. Diffraction data for Dri1-His21Ala +heme and Dri1-His79Ala-Arg90Ala +heme were collected from a single crystal at the 17-ID−1 AMX Beamline (National Synchrotron Light Source II; Brookhaven National Laboratory, Upton, NY), processed via the autoPROC pipeline[68], and anisotropically truncated by STARANISO[69] due to anisotropic diffraction. High-resolution cut-offs were evaluated using STARANISO, CC$_{1/2}$, and $I$ / $\sigma I$ (Supplementary Table 1). The initial phases were determined by molecular replacement using Phaser in Phenix with the WT monomer structure (PDB 8GDW) as the search model. Ligand restraints for HEB were generated using eLBOW[70]. Model refinement was completed with Phenix[71] and model building in Coot[65] (Supplementary Table 1). OMIT difference maps (mF$_{obs}$−DF$_{calc}$) were calculated using Polder, excluding the heme cofactor, His residues, and bulk solvent in the OMIT region[72] (Supplementary Fig. 12).

For Dri1-His79Ala-Arg90Ala +heme, crystal hits were discovered using the Crystal Screen HT (Hampton Research) and Wizard Classic I/II (Rigaku, Mitegen) screens. Crystals were optimized by sitting drop vapor diffusion using an Intelli-Plate 96-3 (Hampton Research) and well solutions of 0.1 M sodium acetate, pH 4.6 – 5.5, 0.2 M NaCl, and 0.75–1.5 M ammonium sulfate. Crystals appeared overnight and were grown for a week before being collected and flash frozen. Diffraction data were obtained and processed similar to described above for Dri1-His21Ala-Heme. Significant translational non-crystallographic symmetry (TNCS) was detected by Xtriage[73]. The largest Patterson peak is 78 Å from the origin at fractional coordinate position (0.5, 0.5, 0.3) with 72% of the origin-peak height. Owing to translational non-crystallographic symmetry issues and merohedral twinning, the R factors are high ($R_{work}$ / $R_{free}$ = 0.311 / 0.326) and the structure was solved at a lower space-group symmetry (P4$_3$) with the twin law h,-k,l (Supplementary Table 1). Chimera[74] and PyMOL[75] were used to visualize and align the structures, measure distances, and prepare figures.

## Small angle X-ray scattering (SAXS)
In-line size-exclusion chromatography coupled to small-angle X-ray scattering (SEC-SAXS) experiments were conducted at the 16-ID (LiX) Beamline (National Synchrotron Light Source II; Brookhaven National Laboratory, Upton, NY). Up to 100 µL of ~1 mM samples were injected onto either a Superdex 200 Increase 5/150 GL column (Cytiva) or a Biozen 3 µM dSEC-2 LC column (200 Å, 300 × 4.6 mm; Phenomenex) and isocratically eluted at 0.35 – 0.5 mL/min using a Shimadzu/Agilent HPLC system with 50 mM HEPES, pH 7.5, 200 mM NaCl degassed buffer at 4 °C. Flow was split 2:1 between the X-ray scattering flow cell and UV-Vis/refractive index detectors. Scattering was collected using Pilatus3X 1 M and 900 K detectors at a sample-detector distance of 3.7 m and at a wavelength of λ = 0.08172 nm, with 2 second exposures per frame. Data were integrated and processed using LiXTools (https://github.com/NSLS-II-LIX/lixtools), py4xs (https://github.com/NSLS-II-

LIX/py4xs), and Jupyter lab[76]. $R_g$ and $P(r)$ analyses were done using BioXTAS RAW[77] and GNOM[78], and molecular weights were calculated by the Bayesian inference method[79]. Electron density maps were generated from averaging and refining 20 runs of DENSS[26] on scattering profiles rebinned to a nonuniform-step $q$-grid to reduce the noise in the low $q$ range. Gaussian smoothing was applied using PyMOL to all maps to aid visualization of the outermost contour. The volume threshold was set at $1.7 \times 10000$ Å$^3$. SAXS-MD-optimized structures and crystal structures were superposed onto electron density maps with DENSS and PyMOL[75]. Data sets were deposited to SASBDB[80]. SAXS data acquisition, sample details, data analysis, modelling fitting and software used are described in Supplementary Table 3.

## Molecular dynamic simulations

The variants of the monomer and dimer structures (His16Ala, His79Ala, His21Ala, His16Ala-His79Ala, His21Ala-His79Ala, His16Ala-His21A, and His79Ala-Arg90Ala) were generated from the WT Dri1 crystal structure, using the mutagenesis wizard in PyMOL[75]. Dri1 and variant structures were used to perform molecular dynamics (MD) simulations using Amber forcefields (Amber 94)[81] to assess the conformational space of proteins in GROMACS (version 2021)[82] which had forcefield parameters available for heme. The starting structures were solved using the TIP3P water model as recommended as the best water model compatible with Amber forcefields (number of water molecules for each dimer structure listed in Supplementary Table 4)[83], and the system was neutralized by adding ions to a concentration of 0.20 M NaCl (Supplementary Table 4). An equilibration simulation was conducted after energy minimization, at constant temperature (300 K, using Nose-Hover[84,85] thermostat with a coupling time (tau-t) of 0.1 ps) and pressure (1 atm, using Berendsen barostat[86] with a coupling time (tau-p) of 1 ps) for 100 ps each. A production simulation for 100 ns was then conducted post equilibration. LINCS algorithm[87] was used to constrain all bonds with an integration time step of 0.002 ps. Lennard-Jones potentials were used to describe short-ranged repulsive and dispersion interactions with a cut off at 10 Å. The particle-mesh Ewald method[88] was used for non-bonded electrostatic calculations with a real space cutoff of 10 Å. The conformation changes observed for both Dri1 and variants were assessed by superimposing the first and last frame obtained from the 100 ns trajectory of the production free-MD simulations. The changes in the position of protein chains and heme (angles and displacement) with respect to the Dri1 structure were assessed using the orientation module in PyMOL[75]. A molecular dynamics simulations checklist is available in Supplementary Information (Supplementary Table 5).

## SAXS-driven MD simulations

The SAXS data for Dri1 and variants (His21Ala and His79Ala-Arg90Ala) of the monomer and dimer proteins and the energy minimized structures from free-MD simulations (described above) were used to perform SAXS-driven MD simulations using a modified version of GROMACS (GROMACS-SWAXS (https://gitlab.com/cbjh/gromacs-swaxs)[89,90]). The SAXS data were converted to energy potential terms to guide the conformation of the protein structures that best explain the experimental data. The starting conformations for the SAXS driven MD for Dri1 were selected as the first and last frame of the free-MD simulation of the Dri1 crystal structure. For the variants, three starting structures were used to perform SAXS-driven MD: a) in silico mutated Dri1 crystal structure, b) last frame (100 ns) of the Dri1-mutated structure from free-MD simulation, and c) experimentally solved crystal structure of the variant. Selecting different starting conformations allowed us to evaluate which orientation of the heme best fits the SAXS data. The simulation parameter file for conducting SAXS-driven MD was obtained from https://cbjh.gitlab.io/gromacs-swaxs-docs/tutorials.html, Byrnes et al. (2023) and Chatzimagas & Hub (2023)[91,92]. Briefly, the force constant (waxs-fc) was set to 1 for Bayesian inference, the memory time used (waxs-tau) was 250 ps, and 30 q points were coupled to the data where the largest q point was set to 10 nm$^{-1}$. The smallest q point was selected based on the experimental data for each case (i.e., 0.06 nm$^{-1}$ for Dri1 and His79Ala-Arg90Ala, 0.08 nm$^{-1}$ for His21Ala). The SAXS-derived force was turned on gradually by setting the waxs-t-target to 5 ns, with relative solvent uncertainty set to 0.1%. The on-the-fly curve calculated at the end of the SAXS-driven MD simulation was scaled to the experimental curve.

Trajectories from the 30 ns SAXS-driven MD simulations for each case were then subjected to the rerun module of GROMACS-SWAXS where an average SAXS intensity curve for the selected time frame was calculated (waxs-tau = −1). The average intensity curve was calculated between 5 ns and 30 ns by setting the GMX_WAXS_BEGIN parameter to 5 ns corresponding to the waxs-t-target used during the SAXS-driven MD. Selecting the simulation beyond waxs-t-target ensures that the conformations obtained have been derived from complete incorporation of SAXS experimental constraints as well as forcefield applied. The average curve obtained after rerunning the trajectories were analyzed to calculate the $R_g$ value and determine if they coincided with the experimentally determined $R_g$ value for each case. The conformations obtained from the extracted trajectories were then tested to fit the experimental data using the FoXS server[93,94]. This additional analysis was done to compare the conformations obtained from SAXS-driven MD and the crystal structures against the SAXS data. Between five and ten conformations were selected for each protein based on the $\chi^2$ values so that the models were at least ten frames apart in the overall SAXS-drive MD trajectory. A molecular dynamics simulations checklist is available in Supplementary Information (Supplementary Table 5).

## Continuous-wave X-band electron paramagnetic resonance (CW-EPR)

CW-EPR spectra were collected at ACERT (Cornell University, Ithaca, NY, USA) with a Bruker EleXSys II spectrometer at 9.39 GHz and at 10 K. Spectrometer settings were as follows: 100 kHz modulation frequency, 6 Gauss modulation amplitude, microwave power of 0.063 mW, and power attenuation of 35 dB. All protein samples were prepared with hemin as described above, except for Zn$^{2+}$-treated samples, which were prepared with excess cofactors (1 molar equivalent hemin and 5 molar equivalents metal) before buffer exchanging to remove unbound species. All samples were exchanged into 100 mM sodium phosphate buffer, pH 7.5, and 20-25% glycerol (w/v) was added as cryoprotectant. Concentrated samples (~1–1.5 mM, ~100 μL) were degassed overnight in an anaerobic box to minimize dissolved oxygen signals and then loaded into quartz tubes (Bel-Art, 660000014; 4 mm O.D.). Spectra were baseline corrected, normalized to the protein concentrations, and analyzed using MATLAB R2019a (MATLAB, MathWorks, USA) and EasySpin[95].

## X-ray Absorption Spectroscopy (XAS)

Dri1 and apo-Dri1 samples were prepared with hemin in 250 mM sodium phosphate buffer, pH 7.5, with 25% glycerol as cryoprotectant, loaded into XAS cells sealed with 25 μm Kapton tape, and flash frozen in liquid nitrogen. X-ray absorption spectra were collected at the Stanford Synchrotron Radiation Lightsource on beamline 9-3, a 2 T 20-pole wiggler side station. Fluorescence data were collected using a monolithic 100-element Ge detector (Canberra). Energy selection was provided by a double Si(220) crystal monochromater oriented to φ = 0, and harmonic rejection was provided by a (spherically bent) Rh-coated mirror. Unwanted signal from photon scattering was reduced using a manganese filter and Soller slits. During data collection, samples were maintained at a temperature of 10 K using an Oxford Instruments CF 1208 liquid helium cryostat. Data were collected at the Fe K-edges, with an in-line Fe foil standard used for energy calibration. Spectra were collected at multiple spots per sample to minimize beam-induced damage. Spectra collected on the same spot were

compared for signs of photodamage prior to later averaging and data processing.

Due to the dilute nature of the samples, and a large background signal from ice diffraction observed in some of the channels, the fluorescence data were analysed channel-by-channel using Larch[96] to identify and exclude the corrupted channels. Channels that were found to be satisfactory were then summed for each scan and imported to Athena[97] for calibration and normalization. Spectra of the reference foil were calibrated to 7111.2 eV, and the energy shift was then applied to the corresponding sample spectrum. The normalized data were then transferred to Pyspline[98] where the EXAFS spectra were extracted by fitting a 4-region spline to the data with polynomial orders 2, 3, 3, and 3. Least-squares fitting of the EXAFS data was done in Artemis[97]. Feff8[99] was used to generate theoretical EXAFS phase and amplitude parameters from an 8 Å radial excision around the central heme-Fe of Dri1 crystal structure. Four parameters were evaluated for each backscattering interaction in course of fitting: the interatomic distance R, the interatomic disorder $\sigma^2$, an energy shift $E_0$, and the amplitude reduction factor $S_0^2$. Of these four parameters, R and $\sigma^2$ were allowed to float for each interaction, while $E_0$ was allowed to float but fixed to a common value among all interactions in a given fit, and $S_0^2$ was fixed to 1.0 for all interactions. The coordination number for each interaction was systematically varied between fits when appropriate within bounds of the expected coordination number from the crystal structure.

### AlphaFold2 complex prediction of SDH

Amino acid sequences of SdhB1, SdhA, and HdrB were used in the AlphaFold2 complex Google Colab[100] (https://colab.research.google.com/github/sokrypton/ColabFold/blob/main/AlphaFold2.ipynb). Unpaired multiple sequence alignment against genetic databases was performed using the MMseqs2 method and unpaired + paired mode. The generated protein complex models were analyzed on ChimeraX software[74].

### Dri1 molecular docking with SdhB1 and interface residues prediction

The crystal structure of Dri1 (UPID: P73129) monomer and Alphafold2 predicted structure of SdhB1 (UPID: P73723) were used to perform molecular docking to identify probable interaction sites between the two proteins. Docking was performed using HEX software version 8.0.0[101,102] to identify the shape and electrostatic compatibility-based interaction interface. The modelled complex structure was energy minimized using the DARS energy minimization protocol. The interaction interface residues were then analyzed by visualizing the docked complex in ChimeraX[74].

### Yeast 2-hybrid assays (Y2H)

The pGBKT7 bait plasmids containing *dri1* or *dri1* mutants were co-transformed into the yeast strain Y2H-Gold (Takara) with the pGADT7-AD prey vectors containing *sdhB1*, *sdhB1* mutants, *sdhB2*, *sdhA*, *hdrB*, *hemA*, *hemB*, *hemL*, *slr0665*, *slr1020*, or *sll1946*, and clones selected on SD -Leu -Trp. Three independent clones were used per interaction and grown in SD -Leu -Trp until saturation. Five microliters were spotted on agar-solidified SD -Leu -Trp (SD - LW) and SD -Leu -Trp -His -Ala (SD −LWHA), medium. Plates were incubated at 30 °C and imaged at 3 days postinoculation.

### Protein pull-down

WT cells expressing Dri1-Strep-tag II, SdhB1-FLAG protein fusions, or untagged proteins were grown in liquid BG−11 at 30 °C and shaken at 90 rpm in an Innova® 44/44 R shaker (New Brunswick) for 6 days. Cell lysis was performed as mentioned above with the addition of 150 mM NaCl. The protein concentrations of the soluble fractions were normalized by Bradford assay (BioRad) for protein pull-downs. A total of 20 μg of soluble proteins were incubated with 10 μL of Strep-Tactin® Sepharose® resin (IBA-LifeSciences) at 24 °C for 2 hours with or without 200 μM hemin. To remove protein unspecific binding, the resin was washed five times with 100 mM Tris-HCl, 150 mM NaCl, 20% glycerol, and 1 mM EDTA pH 8.0. Proteins were eluted with 20 μL of 100 mM Tris-HCl, 150 mM NaCl, 1 mM EDTA, and 2.5 mM desthiobiotin, pH 8.0, boiled for 5 min with SDS-sample buffer, and separated by SDS-PAGE (4-20% ExpressPlus™ PAGE Gel, Genscript), followed by immunoblotting.

### Immunoblot

Separated proteins on PAGE gels were transferred onto PVDF membranes (0.2 μm) using a Trans-Blot Turbo RTA Transfer Kit (Bio-Rad). Blots were washed for 1 min with TBST followed by blocking with Superblock (Thermo-Fisher) for 30 min. Afterwards, blots were incubated overnight at 4 °C with primary antibodies anti-Strep-tag II (1:2000; NBP2-43735 Novus biologicals) or anti-FLAG (1:2000; F1804 Sigma). Unbound antibodies were removed via three 5 min washes with TBST followed by incubation of the blot for 1 h with the HRP-conjugated anti-mouse secondary antibodies (1:10,000; A9044, Sigma-Aldrich). After three successive washes with TBST and one wash with TBS, Thermo Scientific™ SuperSignal™ West Pico PLUS Chemiluminescent Substrate ECL substrate (Thermo) was added, and signal detection was obtained with an ImageQuant™ LAS 4000 (Amersham).

### Confocal microscopy

Subcellular localization of Dri1-eYFP fusion was analyzed in *Synechocystis*. WT cells overexpressing the fusion were grown in liquid BG-11 at 30 °C and shaken at 90 rpm in an Innova® 44/44 R shaker (New Brunswick). YFP fluorescence and chlorophyll autofluorescence were examined using a Leica TCS SP5 confocal microscope. YFP was detected using a 488-nm laser with a detection wavelength window of 500−530 nm and a detector gain of 800, while chlorophyll autofluorescence was detected using an excitation wavelength of 561 nm, a detection wavelength window between 565-615 nm, and a detector gain of 750. Leica Application Suite LAS X software (Leica Microsystems) was used to process the images.

### *Synechocystis* transformation

*Synechocystis* cells were grown in liquid BG−11 at 30 °C and shaken at 90 rpm in an Innova® 44/44 R shaker (New Brunswick) for 6 days. Exponentially growing cells (OD 750 nm ~ 0.5) were harvested at 700 x g for 10 min at 4 °C. Cell pellets were washed three times with BG−11 and finally resuspended in half of the culture volume. Overnight grown *Escherichia coli* DH5α cells containing either pRL443, pRL623, or pCpf1b plasmids were harvested similarly, washed three times in LB lacking antibiotics, and finally resuspended in half of the culture volume. Conjugation of editing plasmids into the different bacterial strains was carried out by mixing 100 μL of *E. coli* pRL443, 100 μL of *E. coli* pRL623, 100 μL of *E. coli* pCpf1b, and 300 μL of *Synechocystis sp.* PCC 6803. Cell mixtures were incubated at 24 °C for 2 h, spread out on a 0.45 μm MF-Millipore membrane filters (Sigma) overlaid on BG−11 agar supplemented with 5% LB, and incubated for 24 h at 24 °C. Filters were transferred onto BG−11 agar supplemented with 50 μg/mL kanamycin. Colonies were transferred to Kan50-BG−11 agar and PCR verified for gene deletion of nucleotide mutagenesis. After Sanger sequencing, correct transformants were grown for 6 days in antibiotic free BG−11 to cure the pCpf1b plasmid, and cured strains were selected on BG−11 agar supplemented with 10% sucrose (*sacB* counterselection gene from pCpf1b).

### Growth of *Synechocystis* mutants and conditions

*Synechocystis* WT and mutated strains were pre-grown in 1 mL of BG−11 media in 24-well plates. After 6 days of growth at 30 °C at 200 rpm in an Innova® 44/44 R shaker (New Brunswick), the cell density

of exponentially growing cells was normalized to an optical density (OD750 nm) of 0.3 (measured in a 96-well plate (non-treated polystyrene, Grenier Bio-One, Monroe, NC, on a Tecan Infinite M1000 Pro microplate reader)). Ten-fold serial dilutions were made using BG−11, and a volume of 5 μL of dilutions $10^0$, $10^{-1}$, and $10^{-2}$ were spotted onto BG−11 agar-solidified media with or without 5 mM glucose and 5 mM glucose + 15 μM DCMU (3-(3,4-dichlorophenyl)−1,1-dimethylurea). Plates were incubated at 30 °C and imaged at 6 days postinoculation. For liquid cultures, cells were diluted to a starting OD750 nm of 0.1 in 1 mL of BG−11 media in 24-well plates and grown at 30 °C at 200 rpm. Cell growth was monitored by OD750 nm for 6 days.

## Chlorophyll fluorescence measurements

WT parental strain and Δ*dri1* were grown in liquid BG−11 at 30 °C with shaking at 90 rpm in an Innova® 44/44 R shaker (New Brunswick). Log-phase cells were harvested and normalized to a chlorophyll concentration of 5 μg mL$^{-1}$ (quantified spectroscopically after methanol extraction[103]). One milliliter of chlorophyll-normalized cells was dark acclimated for 30 min and chlorophyll fluorescence measurements were performed on a Dual-PAM 100 (Waltz), to determine chlorophyll fluorescence parameters ($F_0'$, $F_m'_{dark}$) in the dark. The samples were kept in the dark with a weak measuring light to determine the minimum fluorescence in the dark $F_0'$ for 10 s. A 600 ms saturating light pulse (peak emission at 620 nm, 5,000 μmol photons m$^{-2}$ s$^{-1}$) was applied to determine the dark-acclimated maximal fluorescence $F_m'_{dark}$. Samples were then exposed to actinic light at 50 μmol photons m$^{-2}$ s$^{-1}$ for 5 min. Saturating light pulses were triggered after 30 s, 1 min, 2 min and 5 min after actinic light was turned on. Actinic light was turned off and fluorescence measured for 2 min, with saturating pulses at 30 s, 1 min and 2 min. Data were visualized using the software Dual PAM v1.19. The abundance of major photosynthetic membrane complexes (PSI and PSII) was evaluated with BN-PAGE.

## BN-PAGE

WT parental strain and Δ*dri1* were grown in liquid BG−11 at 30 °C and shaken at 90 rpm in an Innova® 44/44 R shaker (New Brunswick) for 6 days. Exponentially growing cells were harvested at 700 x *g* for 10 min at 4 °C. Cell pellets were resuspended in 750 μL of lysis buffer containing 10 mM HEPES-NaOH pH 7.5, 5 mM NaPO$_4$, 10 mM MgCl$_2$, 10 mM NaCl, 25% glycerol, and Roche cOmplete™ Mini EDTA-free Protease Inhibitor Cocktail (Sigma), and then transferred to Power-Bead Tubes (Qiagen) containing 0.1 mm ceramic beads. Cells were lysed by 10 cycles of 1 min at 30 Hz on a TissueLyserII (Qiagen) separated by 1 min on ice. Unbroken cells were discarded by centrifugation for 5 min at 5000 x *g* at 4 °C. To separate the soluble fraction from the membranes, the supernatant was transferred to a clean 1.5 mL tube and further centrifuged for 30 min at 16,000 x *g* at 4 °C. The supernatant contains the soluble proteins, while the pellet consists of the membranes. The membrane fraction was washed twice with a buffer containing 330 mM sorbitol, 50 mM Bis-Tris pH 7, and Roche cOmplete™ Mini EDTA-free Protease Inhibitor Cocktail (Sigma), and centrifuged for 20 min at 16,000 x *g* at 4 °C. Membranes were resuspended in 100 μL of a buffer containing 20% glycerol, 25 mM Bis-Tris pH 7, 10 mM MgCl$_2$, and 0.1 U DNase, Roche cOmplete™ Mini EDTA-free Protease Inhibitor Cocktail (Sigma), and kept on ice for 20 min. To solubilize the membranes, 10 μL of 10% DDM (n-dodecyl β-D-maltoside) were added to achieve a final concentration of 1% DDM, and tubes were kept on ice for 40 min. A final centrifugation of 30 min at 16,000 x *g* at 4 °C was performed to remove insoluble material. Protein concentration of the solubilized membrane fraction was normalized between samples by DC protein assay (BioRad). A total of 40 μg of membrane proteins were loaded onto a blue-native gel (BN-PAGE, 4-20% ExpressPlus™ PAGE Gel, Genscript).

## Plastoquinone (PQ) pool redox state

PQ pool redox state was indirectly monitored by measuring the fluorescence yield using a dual modulation kinetic fluorometer (model FL-200, Photon Systems Instruments, Brno, Czech Republic), at room temperature[104]. WT and mutant cells were grown at 30 °C under constant illumination (50 μmol) in BG−11 until mid-log phase (OD750 nm between 0.3 and 0.5) and were harvested by centrifugation at 5000 x g for 5 min. Cells were washed once and resuspended in fresh BG−11 at a chlorophyll *a* concentration of 5 μg mL$^{-1}$ (quantified spectroscopically after methanol extraction[103]). The resuspended cells were kept under constant illumination (50 μmol) and at 30 °C with shaking to avoid settling of the cells. Fluorescence measurements were performed with blue measuring flashes (435 nm) of 5 μs. The fluorescence level ($F_0$) was measured for 15 s with measuring flashes of 1 s intervals, followed by addition of 1 mM KCN to block cytochrome oxidase activity. Fluorescence was measured every 10 s for a 6 min time course. The measuring flash itself did not have a noticeable actinic effect and did not trigger PSI and PSII. Actinic light was off during the assay.

## Succinate dehydrogenase activity

SDH activity was measured from isolated membranes of *Synechocystis* cells as detailed in the BN-PAGE section above. A total of 20 μg of membrane proteins were used per 100 μL reaction buffer containing 50 mM potassium phosphate, 0.1 mM EDTA pH 8.0, 5 mM KCN, 120 μM DCPIP, and Roche cOmplete™ Mini EDTA-free Protease Inhibitor Cocktail (Sigma). The reaction was initiated by the addition of 20 mM of succinate, and reduction of the electron acceptor DCPIP was recorded at an absorbance of 600 nm over a time course of 3 h. DCPIP reduction activity was calculated using an extinction coefficient at 600 nm of ε = 21 mM$^{-1}$ cm$^{-1}$.

## Reporting summary

Further information on research design is available in the Nature Portfolio Reporting Summary linked to this article.

## Data availability

The crystallographic data generated in this study have been deposited in the PDB, under accession codes 8GDW (Dri1), 8GF4 (Dri1 + heme), 8FM6 (Dri1 His21Ala + heme) and 8GBK (Dri1 His79Ala-Arg90Ala + heme). The SAXS data generated in this study have been deposited in the SASBDB, under accession codes SASDRH5 (Dri1), SASDRJ5 (Dri1 + heme), SASDRM5 (Co-Dri1 + heme), SASDRK5 (Zn-Dri1), SASDRL5 (Zn-Dri1 + heme), SASDRF5 (Dri1 His21Ala), SASDQS9 (Dri1 His21Ala + heme), SASDRE5 (Dri1 His79Ala-Arg90Ala) and SASDRD5 (Dri1 His79Ala-Arg90Ala + heme). For molecular dynamics simulations, the initial coordinate and simulation input files and a coordinate file of the final output have been deposited on zenodo under the accession code 10789761. Source data are provided with this paper.

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

## Acknowledgements

This work was supported by the U.S. Department of Energy, Office of Science, Office of Biological and Environmental Research, as part of the Quantitative Plant Science Initiative SFA at Brookhaven National Laboratory. E.F.Y. and K.C. are supported by Brookhaven National Laboratory LDRD (21-038). The work (Award DOI 10.46936/10.25585/60000750) conducted by the U.S. Department of Energy Joint Genome Institute (https://ror.org/04xm1d337), a DOE Office of Science User Facility, is supported by the Office of Science of the U.S. Department of Energy operated under Contract No. DE-AC02-05CH11231. This research used beamlines 16-ID, 17-ID–1, and 17-ID-2 of the National Synchrotron Light Source II, a U.S. Department of Energy (DOE) Office of Science User Facility operated for the DOE Office of Science by Brookhaven National Laboratory under Contract No. DE-SC0012704. The Center for BioMolecular Structure (CBMS) is primarily supported by the National Institutes of Health, National Institute of General Medical Sciences (NIGMS) through a Center Core P30 Grant (P30GM133893), and by the DOE Office of Biological and Environmental Research (KP1605010). Use of the Stanford Synchrotron Radiation Lightsource, SLAC National Accelerator Laboratory, is supported by the U.S. Department of Energy, Office of Science, Office of Basic Energy Sciences under Contract No. DE-AC02-76SF00515. The SSRL Structural Molecular Biology Program is supported by the DOE Office of Biological and Environmental Research, and by the National Institutes of Health, National Institute of General Medical Sciences (P30GM133894). This research was supported by the U.S. Department of Energy, Office of Science, Office of Basic Energy Sciences, Chemical Sciences, Geosciences, and Biosciences Division, Physical Biosciences Program through FWP 100593. The contents of this publication are solely the responsibility of the authors and do not necessarily represent the official views of NIGMS or NIH. CW-EPR experiments were run at the National Biomedical Resource for Advanced ESR Spectroscopy (ACERT) at Cornell University, under NIGMS grant 1R24GM146107. We thank Jack H. Freed, Siddarth Chandrasekaran, and Alex L. Lai for their assistance in performing these experiments. The work conducted at Washington University in St Louis was supported by the U.S. Department of Energy (DOE), Office of Basic Energy Sciences, grant DE-FG02-99ER20350 to H.B.P. M.I. was supported by the U.S. Department of Energy, Office of Science, Basic Energy Sciences, Chemical Sciences, Geosciences, and Biosciences Division under field work proposal 449B. K.K.N. is an investigator of the Howard Hughes Medical Institute. We are grateful to Robert Evans from the Joint Genome Institute for the initial cloning of *dri1* into the pET11e vector. We also thank Aditi Bhat (Brookhaven National Lab, Biology dept) for her assistance with ICP-MS measurements, and Kassandra Santiago for her aid in preparing cultures. Crystallization screening of the His21Ala-Heme dimer (PDB 8FM6) was accomplished at the National Crystallization Center at HWI and supported through NIH grant R24GM141256.

## Author contributions

C.E.B.-H. conceptualized the study and performed the bioinformatic analyses. Experiments were designed by N.G., I.K.B. and C.E.B.-H. A.G. and S.C.A. performed the solubility and expression screens that identified ssr1698 as a candidate for crystallization. C.E.B-H., N.G. and E.F.Y. designed in vitro experiments; E.F.Y., D.K. and N.G. performed protein purifications and in vitro experiments. D.K. obtained and solved the crystal structures of wild-type Dri1, and E.F.Y. determined the structures of the variants. D.F.K. collected data of Dri1 mutant crystals. J-F.C. and I.K.B. designed pET11e and HTP cloning strategy. L.Y. and E.F.Y. designed SAXS experiments; J.B. and E.F.Y. collected SAXS data. K.C. and H.V.D performed molecular dynamics simulations. M.A. and R.S. designed and performed the XAFS experiments. N.G. designed and performed in vivo experiments. M.I. and K.K.N. assisted with the chlorophyll fluorescence kinetics analysis. S.B. and H.P. designed and performed the PQ pool reduction kinetics. N.G., C.E.B.-H. and E.F.Y. wrote the manuscript with text and comments from all authors.

## Competing interests

The authors declare no competing interests.
