## [Peer Review File · Nature Communications]

A hemoprotein with a zinc-mirror heme site ties heme availability to carbon metabolism in cyanobacteriaReviewers' comments:

Reviewer #1 (Remarks to the Author):

Grosjean et al present an interesting study on the identification and in-depth characterization of heme protein involved in carbon metabolism. Overall, this is an interesting study that deserves the attention of the Nature Communication readership. I was requested to review the MD simulations and the SAXS-driven MD simulations.

Overall, the combination of MD simulations with SAXS is innovative and provides rigorous means for deriving the fine details of the solution conformations. Prior to publication, several technical details and citations should be added or clarified:

1) Technical details of the MD simulations are missing: the exact force field and water model, GROMACS version, size of the simulation system (number of water molecules), use of salt or use of only counter ions, Lennard-Jones cutoff settings, P- and T-coupling algorithms, use of constraints, equilibration protocol.

2) Citation for the MD simulations are missing (force field, water model, constraint algorithm etc.)

3) Likewise, citations of the SAXS-driven MD simulations are missing (DOI 10.1016/j.bpj.2015.03.062 and 10.1016/j.bpj.2014.06.006)

4) More technical details on the SAXS-driven MD simulations should be added. These correspond essentially to the key parameters provided in the MDP input file:

* force constant (waxsf-c)

* Which memory time did you use (waxsf-tau)?

* How many q-points were coupled to the data, which q-range was coupled to the data?

* How many q-vectors per absolute value of q were used to carry out the orientational average (waxsf-nsphere)?

* Were SAXS-derived forces turned on gradually? If so, over which time? (waxsf-t-target)

* During SAXS-driven MD, did you adjust the scale and an offset between simulation and data before computing SAXS-derived forces (waxsf-lexp-fit)?

* Which relative uncertainty of the solvent density was assumed (waxsolvdens-uncert)? This parameter may reduce the weights of q-points at very small angles, where the experimental uncertainties are often spuriously small, while the experimental $I(q)$ curve may be subject to (minor) systematic errors.

5) The error bars of the upper panel of Supplementary Fig 3a-c seem very large. Note that, the "Final SAXS intensity" is a comparison of only the final on-the-fly calculated SAXS curve (involving only an average over the final memory given by waxstau) with the data. For proper comparison, I strongly suggest using the "rerun" functionality of the Gromacs mdrun module and using a uniform time average (with waxstau=-1). For instance, you could compute the SAXS curve from an average over the last 5ns of the SAXS-driven MD simulation, which would give much lower error bars and more rigorous comparison between SAXS-driven MD simulation and the data. This way, it would also not be required to use FoXS for the final comparison. Note that FoXS uses an approximate treatment of the hydration layer and excluded solvent and uses several fitting parameters, so you may overlook some finer details in the detail (which had been captured before in the explicit-solvent SAXS calculation used by GROMACS-SWAXS).

Reviewer #2 (Remarks to the Author):

The most significant result of the manuscript, presented by C. E. Blaby-Haas et al., is the discovery of a so far unknown family of dimeric hemoproteins, in which the heme Fe atom is axially coordinated by two Zn atoms, via nearby histidine residues and thus forming a two-fold symmetric Zn-His-Fe-His-Zn site, which is thoroughly characterized by different structure-sensitive techniques, including Extended X-ray Absorption Fine Structure (EXAFS) spectroscopy.

The experimental results using EXAFS on two different samples, Dri 1 and apo-Dri 1, are summarized in Fig. 3 f-g (main text) and Fig. S4 (supplementary material). Standard EXAFS analysis approach was applied to both samples and extracted fit results are presented in tables included in Fig. 3g and Fig. S4b-c. The authors used the two structural models to fit the EXAFS data of the Dri 1 sample using 6-fold coordination with N atoms in the first coordination shell around Fe: in one case a single six-coordinate N scattering path was used (Fig. S4b), and an axially distorted structure with 4 shorter equatorial Fe-N bonds (heme ring) and additional 2 longer axial Fe-N bonds (His), which further coordinated to Zn atoms at longer distances from Fe atom. As a control sample, apo-Dri 1 was used and measured under the same experimental conditions. These EXAFS results are summarized in Fig. S4c. In this case no Zn atoms were used in the model and the Fe-N coordination is modeled with 5 Fe-N shorter bonds and one distant N/O bond.

The procedure of EXAFS data reduction and analysis is sufficiently explained in the manuscript and figure captions and the presented fit results are sufficient to draw certain conclusions from the fitted models.

However, the main results of EXAFS analysis rely on very subtle differences in the measured data, which should be included or explained in the manuscript better. The reviewer's objections to the presented analysis and some suggestions how to address them in the revised version of the manuscript are added in a separate file.

Reviewer #3 (Remarks to the Author):

In Grosjean et al., the authors report characterization of the a conserved protein domain capable of binding heme and perform a series of structural, phylogenetic, and mutational analyses in order to dissect the role of this protein domain. The authors are particularly focused upon a cyanobacterial protein (named Dri1) uniquely composed of the single conserved protein domain (DUF2470) without other evident protein domains. The authors detail a heme-dependent binding affinity between Dri1 and subunits of the succinate dehydrogenase complex.

As the expertise of this reviewer was sought in the area of cyanobacterial physiology and evaluation of photosynthetic activity, I will restrict my comments to the experiments, claims, and data interpretation associated with these topics. Unfortunately, there are serious concerns about the methodology used to analyze photosynthesis in the variety of Dri1 mutants that can considerably impact the conclusions that can be drawn.

Major comments:

The authors do not sufficiently describe the methodology behind the chlorophyll a fluorescence kinetics and the PQ redox state measurements at a level of detail to fully diagnose the data shown. However, the data appears to show some very unexpected results that may indicate some core methodological error(s). This reviewer suspects that this may be closely related to the author's use of a FluorCam 800MF: this line of instrumentation appears to have been developed for the analysis of plants. There are a number of substantial differences between the photosynthetic machinery between cyanobacteria and plants that require adjustment of the instrumentation and the equations used to evaluate cyanobacterial samples relative to plants when estimating photosynthetic performance. I elaborate upon a few concerns related to this analysis below:

1. The authors don't fully describe how they are standardizing their cyanobacterial samples prior to Chl a fluorescence analysis. It is important to standardize the chlorophyll a content of samples during analysis in order to have a similar baseline. Additionally, because the phycobilins can contribute to F₀ in cyanobacteria due to their overlapping excitation/emission spectra, when there is a large difference in F₀ values seen (as in Figures 6AB), it may be important to do a more detailed pigment analysis to determine if the phycobilisome content or other reactive pigments are substantially altered between the different strains.

2. Something is very unusual with the fluorescence trace provided in Figure 6b. Cyanobacteria are generally in State II (phycobilisomes associated predominantly with PSI) in the dark. Upon turning on actinic light, electrons are pulled from the PQ pool, leading to an overall oxidation of the photosynthetic

electron transport chain and leading to a characteristic rise in Chl_a fluorescence as the cells shift to State I (PBS associated more with PSII). It is notable that this is a distinction between cyanobacterial analysis and plants. The authors show data that seems to describe the reverse (Chl_a fluorescence drops in the light).

a. This gives rise to some confusing data (for instance, the F₀ fluorescence is higher than the F'₀ fluorescence). Indeed, the F'_M (maximum fluorescence in the light) is substantially lower than F₀ (minimal fluorescence in the dark). Similarly the maximal fluorescence under a saturation pulse decreases as the cells are placed into light (as well as the variable fluorescence), a situation that is reversed from expected.

b. It is unclear how the authors are deriving their claims of increased F_v in the Dri1 mutant (claims beginning on line 261). Both the F_v in the dark, and the F'_v (variable fluorescence from F'₀ to F'_M) appear from the trace to be higher in WT. It is somewhat difficult to read these traces at the size given to the reviewers, but at all points along the WT trace (dark green) the size of the induced change in fluorescence via the saturation pulse appears to be greater than in the mutant trace. Regardless, it is somewhat difficult to interpret these results given that the state transition is likely still occurring throughout the entire 60 seconds of actinic light exposure at the actinic light intensity used.

c. While it is appreciated that the authors show the fluorescence across the well as a part of their data in Figures 6a and 6b, it is not easy to explain why there is a substantial difference in fluorescence observed across the sample, as these should be homogenous, liquid suspensions. Were any controls used to rule out artifacts of the use of the 96 well format (this is atypical) and were optically appropriate 96 well plates used?

3. Given the rather serious concerns with the measurements of some key parameters (e.g., F₀) in the above comments, it would be important to provide some additional information on the PQ reduction assay that was performed in panels 6D-6F and to elaborate upon controls. To the knowledge of this reviewer, this is an atypical experiment to perform on cyanobacterial samples.

Given these concerns, the conclusions reached related to Figure 6A-6F may be seriously compromised. There is a considerable amount of additional data in this manuscript that supports other elements of the conclusion and model that is ultimately proposed. However, the photosynthetic measurements (Lines 255-280) and the conclusions related to them (see Discussion beginning on Line 328) should be revisited.

Minor comment:

Elements of the language composition can be difficult to follow at times. For example, in the paragraph beginning on Line 221, there is considerable redundancy in the way that topic points are introduced and discussed. For instance, the section beginning on line 227 about SdhB1 repeatedly switches between HdrB and SdhC/D, repeats elements of the cyanobacterial SdhB1 c-terminal extension and repeats the cyanobacterial HdrB fusion. Streamlining the language may help reader comprehension. In some cases where the discussion of Sdh subunits more broadly (across many prokaryotes and eukaryotes) is in the same discussion as the cyanobacterial-specific Sdh subunits, it may help to tweak the nomenclature so that it is clear what homolog(s) are being discussed.

Reviewer #4 (Remarks to the Author):

1. How was the outer resolution limit determined, using $CC1/2$ or $Mn(I)/sd$? The authors might include $CC1/2$ in the crystallographic statistics table.

2. Line 514-519: The authors described densities of the heme and disordered regions but did not provide any figures. It would be helpful if the authors include representative figures showing density maps around ligands and key residues to illustrate the quality of the maps.

3. Some structures were refined with Refmac5 and others with Phenix. Was this selection random, or were there specific reasons for choosing these refinement programs?

4. The gap between R_{free} and R_{work} can also be attributed to overfitting or an inappropriate resolution limit. Did the authors use different refinement strategies and resolution limits to obtain better R_{free} and R_{free}/R_{work} gap values? Additionally, the real density map should be always inspected to avoid overfitting.

5. All of the spectral figures, such as Supplementary Figure 8, are challenging to read. The colors used are too similar, and the curves appear very thin.

Reviewer #5 (Remarks to the Author):

Comments on "A hemoprotein with a zinc-mirror heme site ties heme availability to carbon metabolism in cyanobacteria".

The manuscript reports identification of a family of dimeric hemoproteins which is suggested to be a heme-binding regulatory domain. Authors suggest renaming of DUF2470 as Domain Related to Iron (DRI). DUF2470 entry in data bank describes it as an uncharacterised domain, found in a group of putative heme-iron utilisation protein. To substantiate this, they have focussed on a cyanobacterial specific protein clade where DRI is not fused to another domain which they have called Dri1 to distinguish it from DRI. Synthesis of heme and its regulation is important topics which makes this study of some interest and potentially worthy of its publication in Nature communications.

Authors have applied a number of experimental approaches as well as utilised computational modelling and predictive tools to support their hypotheses and characterise this interesting protein. Though a variety of techniques have been applied, some of the key data are of limited quality lacking robustness required for such new findings and publication in influential journals such as NCOMMS. As examples (but not limited to) I list some of the weaknesses:

1. There are four crystallographic structures, two for Dri1 (apo and apo+heme) and two for mutants (a single and a double mutation). Given the low molecular weight and a soluble nature of the protein, the data quality is poor, and resolution of the structures is what is considered these days very low. The best resolution is that of Dri1 (2.35Å) but with very high temperature factors. Even though the authors have not provided Wilson B for the data sets, I estimate that it is very high indeed reflecting poor quality of crystals calling for further optimisation. Truncation of data may help but my guess is that it would not be sufficient. This comment is applicable to all of the 4 structures.

2. The combination of SEC-SAXS is good to see together with the latest approach which attempts to calculate electron density from a one-dimensional scattering curve. This approach is still not well-tested and not yet well known – thus a fuller description of strength and limitations should have been incorporated highlighting some of the key points from reference 61. It is important to remember that the electron

density of proteins (~ 0.43 electrons/Å³) is only slightly higher than the average electron density of water (0.33 electrons/Å³) and any subtle changes in the buffer can make a significant difference. This is even more so important for smaller MW proteins. It is unclear in Figure 3(a) if they are of raw scattering curves (corrected for buffer) or smoothed. Curves in supplementary Figure 3 are of raw data but then fits to most of the curves are not particularly good. Non-uniqueness of the SAXS structures need to be taken into account when reporting part of the evidence for a significant advance.

3. Data quality of XAFS is good but some of the interpretation is unclear and may be over-stretched. It has not been made clear if the multiple scattering from the pyrrole rings have been taken into account. Figure 3 should also include a schematic of N-C paths that have been included and the multiple scattering (MS) contributions that have been included. For MS pathways, the angle information should also be included and if these were constrained or allowed to float. The assignments of peaks between 5 and 5.5Å should be handled with care. Scattering strength from such distance even for Zn is limited and their presence is largely in low k range. One way to gain some credibility would be to perform a difference EXAFS analysis i.e. subtracting the fit without Zn from the experimental spectra and then fitting the difference spectrum with Zn. Current EXAFS fit involves too many parameters for the number of independent data points in the limited k-range. The authors claim to have estimated standard deviations for distance on the order of ± 0.02 Å for all of the distances determined from EXAFS analysis. This is incorrect. They are fitting errors from an over-fitting situation – not the accuracy of distances. The first shell distance is accurate and is likely to be accurate to ± 0.02 Å but not others due to presence of multiple scattering, weaker contribution from distant atoms and overlapping contributions from outer

shells that would result in cancellation of signals for part of the k-range. A thorough analysis of χ^2 with the addition of each shell is required to assess the significance of contribution and the uncertainty in their determination.

4. There is very limited evidence (hard data) provided for the interaction between Dri1 and succinate dehydrogenase and the presence of a complex. At least a SAXS evidence for the complex in Figure 5 would have provided some confidence. Also, inability to identify a DRI fusion protein in a cyanobacterial is somewhat weakens the overall hypothesis.

5. The authors suggest an increase in heme-binding capacity as well as an increase in the heme-binding affinity with Zn. It would have been good to have crystallographic structures with and without Zn (but with heme) as it would have allowed to assess 'any optimisation' of heme stereochemistry at the homodimeric interface.

6. 20ns of MD calculations are on lower end of the scale. It is fairly simple to extend these to 100ns.

Minor points:

1. Authors need to clarify that heme binding and dimeric assembly can occur without Zn.

2. Dri1 is mentioned in the abstract without specifying what it stands for.

3. Methods section is too long, some of it could be placed in supplementary section and some can be deleted altogether. Data collection and refinement statistics given between lines 548 and 549 are unnecessary and not normal. Given the supplementary tables 1 and 2, this information should be deleted from the methods section.

4. Figure 4 caption – clarify that panels e to h are from experimental crystallographic structures. For panels c and d, SAXS raw data and fits should be shown.

Reviewer #1:

Grosjean et al present an interesting study on the identification and in-depth characterization of hemeprotein involved in carbon metabolism. Overall, this is an interesting study that deserves the attention of the Nature Communication readership. I was requested to review the MD simulations and the SAXS-driven MD simulations.

Response: Thank you for taking the time to read our manuscript and providing your expert opinion and recommendations for strengthening the manuscript. We have now addressed your comments below and made corresponding edits to the manuscript as indicated.

Overall, the combination of MD simulations with SAXS is innovative and provides rigorous means for deriving the fine details of the solution conformations. Prior to publication, several technical details and citations should be added or clarified:

1) Technical details of the MD simulations are missing: the exact force field and water model, GROMACS version, size of the simulation system (number of water molecules), use of salt or use of only counter ions, Lennard-Jones cutoff settings, P- and T-coupling algorithms, use of constraints, equilibration protocol.

Response: These technical details have now been added to the methodology section describing the MD simulations. It now reads:

Starting at line 487:

"Dri1 and variant structures were used to perform molecular dynamics (MD) simulations using Amber forcefields (Amber 94)⁷³ to assess the conformational space of proteins in GROMACS (version 2021)⁷⁴. The starting structures were solved using the TIP3P water model (~143,078 water molecules)⁷⁵, and the system was neutralized by adding ions to a concentration of 0.20 M NaCl. A 100 ns production simulation after energy minimization was conducted at constant temperature and pressure (300 K and 1 atm) using the particle-mesh Ewald method⁷⁶ for long-range electrostatic interactions with a cutoff of 10 Å for non-bonded calculations."

Citations:

73. <https://doi.org/10.1021/ja00124a002>

74. <https://doi.org/10.5281/zenodo.4457591>

75. <https://doi.org/10.1063/1.445869>

76. <https://doi.org/10.1063/1.464397>

2) Citation for the MD simulations are missing (force field, water model, constraint algorithm etc.)

Response: The citations have been added to the text, please see the pasted text from above, starting at line 487 in MS.

3) Likewise, citations of the SAXS-driven MD simulations are missing (DOI 10.1016/j.bpj.2015.03.062 and 10.1016/j.bpj.2014.06.006)

Response: The link to the modified GROMACS for small-angle scattering calculations in GitLab has been added together with the citations (see pasted text in comment #4 below).

4) More technical details on the SAXS-driven MD simulations should be added. These correspond essentially to the key parameters provided in the MDP input file:

* force constant (waxsf-c)

* Which memory time did you use (waxsf-tau)?

* How many q-points were coupled to the data, which q-range was coupled to the data?

* How many q-vectors per absolute value of q were used to carry out the orientational average (waxsf-nsphere)?

* Were SAXS-derived forces turned on gradually? If so, over which time? (waxs-t-target)

* During SAXS-driven MD, did you adjust the scale and an offset between simulation and data before computing SAXS-derived forces (waxs-lexp-fit)?

* Which relative uncertainty of the solvent density was assumed (waxs-solvdens-uncert)? This parameter may reduce the weights of q-points at very small angles, where the experimental uncertainties are often spuriously small, while the experimental I(q) curve may be subject to (minor) systematic errors.

Response: The MDP file was originally obtained from the GROMACS_SWAXS tutorial example (from <https://cbjh.gitlab.io/gromacs-swaxs-docs/tutorials.html>) and later fine-tuned using the ref (<https://doi.org/10.1016/bs.mie.2022.09.014>). We have added the details of the MDP file briefly in the text. It now reads:

Starting at Line 499:

“The SAXS data for Dri1 and variants (His21Ala and His79Ala-Arg90Ala) of the monomer and dimer proteins and the energy minimized structures from free-MD simulations (described above) were used to perform SAXS-driven MD simulations using a modified version of GROMACS (GROMACS-SWAXS (<https://gitlab.com/cbjh/gromacs-swaxs>)^{77,78}). The SAXS data were converted to energy potential terms to guide the conformation of the protein structures that best explain the experimental data. The starting conformations for the SAXS driven MD for Dri1 were selected as the first and last frame of the free-MD simulation of the Dri1 crystal structure. For the variants, three starting structures were used to perform SAXS-driven MD: a) *in silico* mutated Dri1 crystal structure, b) last frame (100 ns) of the Dri1-mutated structure from free-MD simulation, and c) experimentally solved crystal structure of the variant. Selecting different starting conformations allowed us to evaluate which orientation of the heme best fits the SAXS data. The simulation parameter file for conducting SAXS-driven MD was obtained from <https://cbjh.gitlab.io/gromacs-swaxs-docs/tutorials.html>, Byrnes et al. (2023) and Chatzimagas & Hub (2023)^{79,80}. Briefly, the force constant (waxs-fc) was set to 1 for Bayesian inference, the memory time used (waxs-tau) was 250 ps, and 30 q points were coupled to the data where the largest q point was set to 10 nm. The smallest q point was selected based on the experimental data for each case (i.e., 0.06 nm for Dri1 and His79Ala-Arg90Ala, 0.08 nm for His21Ala). The SAXS-derived force was turned gradually by setting the waxs-t-target to 5 ns, with relative solvent uncertainty set to 0.1%. The on-the-fly curve calculated at the end of the SAXS-driven MD simulation was scaled to the experimental curve.

Trajectories from the 30 ns SAXS-driven MD simulations for each case were then subjected to the rerun module of GROMACS-SWAXS where an average SAXS intensity curve for the selected time frame was calculated (waxs-tau= -1). The average intensity curve was calculated between 5 ns and 30 ns by setting the GMX_WAXS_BEGIN parameter to 5 ns corresponding to the waxs-t-target used during the SAXS-driven MD. Selecting the simulation beyond waxs-t-target ensures that the conformations obtained have been derived from complete incorporation of SAXS experimental constraints as well as forcefield applied. The average curve obtained after rerunning the trajectories were analyzed to calculate the R_g value and determine if they coincided with the experimentally determined R_g value for each case. The conformations obtained from the extracted trajectories were then tested to fit the experimental data using the FoXS server^{81,82}. This additional analysis was done to compare the conformations obtained from SAXS-driven MD and the crystal structures against the SAXS data. Between five and ten conformations were selected for each protein based on the χ^2 values so that the models were at least ten frames apart in the overall SAXS-driven MD trajectory.”

5) The error bars of the upper panel of Supplementary Fig 3a-c seem very large. Note that, the "Final SAXS intensity" is a comparison of only the final on-the-fly calculated SAXS curve (involving only an average over the final memory given by waxs-tau) with the data. For proper comparison, I strongly suggest using the "rerun" functionality of the Gromacs mdrun module and using a uniform time average (with waxs-tau=-1). For instance, you could compute the SAXS curve from an average over the last 5ns of the SAXS-driven MD simulation, which would give much lower error bars and more rigorous comparison between SAXS-driven MD simulation and the data. This way, it would also not be required to use FoXS for the final comparison. Note that FoXS uses an approximate treatment of the hydration layer and excluded solvent and uses several fitting parameters, so you may overlook some finer details in the detail (which had been captured before in the explicit-solvent SAXS calculation used by GROMACS-SWAXS).

Response: The Final SAXS curves were calculated using the rerun module with waxs-tau=-1, however for a smaller time scale ranging from 1.5-2 ns for each case. As described above, we have now extended the simulation to 30 ns and using the rerun module calculated the Average SAXS Intensity plots for time scale 5 ns to 30 ns which have reduced the error bars to a great extent (see Supplementary Fig. 4 and 9). Moreover, we have now also included the SAXS-driven MD simulations of the variants using 3 different starting structures to have a better comparison of the model to data fit (see Supplementary Fig. 4 and 9). The models generated using the crystal structure of the variants fit the best against the SAXS data. The FoXS server was only used to compare the fit obtained from SAXS-driven MD models with the crystal structures which are now included in the Supplementary Fig. 4 and 9.

Reviewer #2:

The most significant result of the manuscript, presented by C. E. Blaby-Haas et al., is the discovery of a so far unknown family of dimeric hemoproteins, in which the heme Fe atom is axially coordinated by two Zn atoms, via nearby histidine residues and thus forming a two-fold symmetric Zn-His-Fe-His-Zn site, which is thoroughly characterized by different structure-sensitive techniques, including Extended X-ray Absorption Fine Structure (EXAFS) spectroscopy.

The experimental results using EXAFS on two different samples, Dri1 and apo-Dri1, are summarized in Fig. 3 f-g (main text) and Fig. S4 (supplementary material). Standard EXAFS analysis approach was applied to both samples and extracted fit results are presented in tables included in Fig. 3g and Fig. S4b-c. The authors used the two structural models to fit the EXAFS data of the Dri1 sample using 6-fold coordination with N atoms in the first coordination shell around Fe: in one case a single six-coordinate N scattering path was used (Fig. S4b), and an axially distorted structure with 4 shorter equatorial Fe-N bonds (heme ring) and additional 2 longer axial Fe-N bonds (His), which a further coordinated to Zn atoms at longer distances from Fe atom. As a control sample, apo-Dri1 was used and measured under the same experimental conditions. These EXAFS results are summarized in Fig. S4c. In this case no Zn atoms were used in the model and the Fe-N coordination is modeled with 5 Fe-N shorter bonds and one distant N/O bond.

The procedure of EXAFS data reduction and analysis is sufficiently explained in the manuscript and figure captions and the presented fit results are sufficient to draw certain conclusions from the fitted models. However, the main results of EXAFS analysis rely on very subtle differences in the measured data, which should be included or explained in the manuscript better. The reviewer's objections to the presented analysis and some suggestions how to address them in the revised version of the manuscript are added in a separate file.

Response: We appreciate the reviewer's recognition of this work and the significance of the role that XAS played in the characterization. Please find our detailed responses to the objections raised below.

Pasted separate PDF file:

Comments to the Authors (EXAFS data analysis)

The structural changes in different models used in EXAFS analysis of both Dri1 and apo-Dri1 samples are sufficiently described to understand how the fit procedure was carried out. However, the main findings of the manuscript rely on comparison of these results to favor one of the models for the Dri1 sample (including an axially distorted first coordination shell with Fe-N bond distances and an additional Zn atoms at large distance from the absorber). It is therefore I have some objections to how the different models were compared to arrive to the conclusions presented in the main text of the manuscript. The details of my suggestions to the authors are outlined below:

- 1) Structural model of the first coordination shell in Dri1 sample: it is concluded that axially distorted model with 4 equatorial Fe-N bonds and 2 axial Fe-bonds fit the data better. However, comparing the statistical goodness of both fits (in the same k-range), the authors obtained very similar R-factors and reduced chi-square χ^2 red for the same k- and R-ranges used in both models. Given the small differences between 6-coordinate N model and 4+2- coordinate N model (in the first case the fit results deliver 6 x Fe-N distances of 2.04 Å and in the second one 4 x Fe-N distances of 2.01 Å and 2 x Fe-N distances of 2.12 Å), a more rigorous statistical comparison of both fits results should be performed.

Response: We agree with the reviewer's point that they are nearly identical in terms of statistical quality. The slightly better fitting statistics of the 4+2 model is almost certainly due to the inclusion of two additional fitting parameters (an additional ΔR and σ^2 value in the first shell). A statistical comparison of the 6 and 4+2 coordinate fits using the Hamilton test confirms that the small decrease in the R-factor incurred in the 4+2 fit is not statistically significant. However, we had decided to include both fits in the manuscript in response to the crystallography data, which supports differing axial and equatorial Fe-N lengths and have now noted in the main text that the EXAFS resolution needed to confidently tease apart these two sub-shells is beyond the resolution of our data set. We have updated the text in the manuscript to better reflect this reasoning.

The sentence beginning on line 168 now reads:

"While the resolution of EXAFS data does not allow high confidence in the distances in this split-coordination fit, it is consistent with the distribution of Fe-N(heme) and Fe-N(His79) distances observed in the crystal structure."

- 2) Here, the authors reported both R-factors and χ^2 red values, however other parameters were not identical in both fits: values of ΔE changed (1.9 eV and 2.5 eV), which in my opinion is a little arbitrary, especially that apo-Dri1 uses a significantly different $\Delta E = -0.8$ eV (difference of 3.3 eV with respect to best-fit value for Dri1). Having said that, and assuming no big impact of ΔE (although it is known it will affect the bond length fit results), the R-factors for both sets of data are nearly identical: 0.026 (Fig. 3g) vs. 0.028 (Fig. S4b). For apo-Dri1, the corresponding R-factor is much lower and indicates a much more statistically reliable fit. 2)

Response: These small differences in ΔE have little impact on the bond distances predicted by the EXAFS models. We repeated the 6 coordinate and 4+2 coordinate fits to the WT data with the value of ΔE fit to 1.9, 2.5 and -0.8. (Table X1-X4). These analyses show that the resulting fits are nearly identical.

WT	Path	Coordination	R(Å) ^a	σ^2 (Å ²) ^b	ΔE	K range	R-factor	χ^2 red				
	N	2	2.12	200	1.9	2-13	0.027	36.9				
	N	4	2.01	349								
	C	2	2.94	234								
	C	8	3.07	234								
	N-C	26	3.26	1206								
	C	4	3.77	351					R range	N_{var}	N_{dep}	
	C	8	4.09	2212								
	N-C	16	4.32	704								
	N-C	24	5.33	1030						1-6	21	34.6
	Zn	1	5.53	335								
	N-C	26	5.69	780								

WT	Path	Coordination	R(Å) ^a	σ^2 (Å ²) ^b	ΔE	K range	R-factor	χ^2 red				
	N	6	2.05	634	2.5	2-13	0.029	34.7				
	C	2	2.94	227								
	C	8	3.07	227								
	N-C	26	3.26	1314								
	C	4	3.77	339					R range	N_{var}	N_{dep}	
	C	8	4.07	1734								
	N-C	16	4.32	743								
	N-C	24	5.34	1214						1-6	19	34.6
	Zn	1	5.54	369								
	N-C	26	5.70	797								

WT	Path	Coordination	R(Å) ^a	σ^2 (Å ²) ^b	ΔE	K range	R-factor	χ^2 red				
	N	2	2.09	164	-0.8	2-13	0.032	44.5				
	N	4	1.99	409								
	C	2	2.93	262								
	C	8	3.06	262								
	N-C	26	3.26	817								
	C	4	3.75	357					R range	N_{var}	N_{dep}	
	C	8	4.24	2047								
	N-C	16	4.31	657								
	N-C	24	5.28	419						1-6	21	34.6
	Zn	1	5.5	263								
	N-C	26	5.64	818								

WT	Path	Coordination	R(Å) ^a	σ^2 (Å ²) ^b	ΔE	K range	R-factor	χ^2 red				
	N	6	2.03	621	-0.8	2-13	0.035	42.3				
	C	2	2.92	257								
	C	8	3.06	257								
	N-C	26	3.26	830								
	C	4	3.75	354					R range	N_{var}	N_{dep}	
	C	8	4.24	2054								
	N-C	16	4.31	657								
	N-C	24	5.28	422						1-6	19	34.6
	Zn	1	5.49	269								
	N-C	26	5.64	825								

- 3) The inclusion of Zn atoms, of the assignment of Zn-related scattering path in Fourier transformed EXAFS data shown in Figs. 3f and S4a is hard to believe unless a control fit results were shown. The authors consider a very weak peak in FT magnitude and very large distance from the absorbing Fe atoms (> 5.5 Å). At such long distances, in frozen solution, EXAFS data is not very reliable and compared to noise level at high R-values.

Response: We have provided a control fit and further comments below in response to point #4. With respect to the reviewer's expression of concern regarding the peak we are designating as arising from the Fe-Zn interaction, we would first like to draw a comparison between our Zn-loaded sample and the EXAFS in these references: <https://doi.org/10.1021/ja043374r>, <https://doi.org/10.1021/bi501489r>, [https://doi.org/10.1016/S0003-9861\(03\)00403-X](https://doi.org/10.1016/S0003-9861(03)00403-X), <https://doi.org/10.1039/D0RA02335C>. The data presented in these studies include EXAFS of heme without a strong scattering atom such as Zn between 5 and 6 Å, and, as such, lack the prominent scattering peak that is observed at 5.2 Å (non-phase corrected) in our data. While we agree with the sentiment that "At such long distances, in frozen solution, EXAFS data is not very reliable and compared to noise level at high R-values", biological EXAFS has been used in the past to fit long-distance metal-metal interactions (e.g., <https://doi.org/10.1073/pnas.1422058112>, <https://doi.org/10.1021/ja00085a022>, <https://doi.org/10.1038/nchem.2055>, <https://doi.org/10.1021/ja00066a019>, [https://doi.org/10.1016/0076-6879\(93\)26028-8](https://doi.org/10.1016/0076-6879(93)26028-8)). In these references, the long metal-metal component is small and is close to the noise level in the FT data. In our case, the comparison of the Zn-unloaded and loaded datasets gives strong support for the presence of a new intensity at 5.2 Å in the non-phase corrected Fourier-transformed EXAFS. In biological systems, such long-distance components can only appear from the presence of a strong backscatter such as a heavy metal.

- 4) Would it be possible to present an EXAFS fit where the identical structural model that has been used in Fig. 3f-g was used to fit the data without Zn atom? What is the statistical significance of having this additional atom

included? Does it improve the R-factor significantly? In my opinion, while EXAFS fit results for apo-Dri1, shown in Fig. S4c, are reliable and statistically significant (R-factor = 0.011), the differences in EXAFS fits for Dri1 using two distinct Fe-N coordination models are difficult to distinguish. The further inclusion of Zn atom to one of the models makes the interpretation even more complicated unless careful Statistical comparison is performed. Since the scattering path related to presence of Zn at large distance from Fe atoms is a very weak contribution to the overall EXAFS signal (signal strength on a similar level to noise), I think it is difficult to provide a solid proof for this finding from the analyzed EXAFS data.

Response: The reviewer is correct to note that the Fe-Zn scattering makes only a small contribution to the EXAFS. Even for a heavier scattering atom like Zn (compared to a more typical atom like C, N or O), the scattering contribution at distances $>5 \text{ \AA}$ is low. To alleviate these concerns, we have repeated both the 6 coordinate and 4+2 coordinate WT EXAFS fits without Zn, and then assessed how the inclusion of the Zn path increases the goodness of fit using the Hamilton test. These fits are now presented in Supplementary Fig. 5g-h. In the limit of only assessing the impact of the additional two variables needed to model the Fe-Zn path (ΔR and σ^2), the exclusion of the Zn path in both the 6 and 4+2 coordinate fits resulted in an increase in the r-factor by 0.04 ($I_r = 0.1$ when assessed with the Hamilton test). We have included the 6 and 4+2 coordinate fits without the Fe-Zn path present in the SI for comparison (Supplementary Fig. 5g-h).

In further support of our designation of the feature at 5.2 \AA (non-phase corrected) as arising primarily due to the Fe-Zn scattering path, we have isolated and Fourier-filtered this peak, and we have plotted with it the simulated EXAFS of an Fe-O, Fe-N-C (multiple scattering) and Fe-Zn backscattering contribution, with the backscattering atom at a distance of 5.6 \AA from the Fe (this difference of $\sim 0.4 \text{ \AA}$ accounts for phase shift correction) (Supplementary Fig. 5d). The variables for these paths (S_0^2 , ΔE , ΔR , and σ^2) were kept the same as they were for the Fe-Zn path in the 4+2 fit. Of the three paths tested, the Fe-O and Fe-N-C are too weak and out of phase to yield the observed peak, whereas the Fe-Zn is comparable in phase and amplitude. To further test this, we repeated this test, but increased the coordination number of the Fe-O and Fe-N-C paths (Supplementary Fig. 5e). These paths (Fe-O and Fe-N-C) are still out of phase with the Fourier-filtered peak and have a significantly different amplitude profile. This figure is now included in the SI and we have updated the main text to include a sentence that reads:

Line 172: "This Fe-Zn path at 5.54 \AA is in reasonable agreement with the crystal structure, supporting the existence of this zinc-mirror binding site (Fig. 3f-g, Supplementary Fig. 5a-c). Other possible scattering interactions that could give rise to this peak were tested (Supplementary Fig. 5a,d-e) and found to be incompatible on the basis of their phase as observed in a comparison with the Fourier-back transform of the peak centered at $\sim 5.3 \text{ \AA}$ in the EXAFS data."

The reviewer also writes that "Since the scattering path related to presence of Zn at large distance from Fe atoms is a very weak contribution to the overall EXAFS signal (signal strength on a similar level to noise), I think it is difficult to provide a solid proof for this finding from the analyzed EXAFS data." We agree with the reviewer. The EXAFS data set presented here are complementary to the crystallographic and EPR data sets that each provide a line of evidence supporting the Zn-binding mediated Fe(III) spin transition. In this case, crystallography supports the Zn binding $\sim 5.5\text{-}6 \text{ \AA}$ from the Fe, while the EPR supports a Zn-mediated spin transition at the Fe(III) in solution samples (Fig. 3h). Likewise, our EXAFS data supports these two data sets by providing evidence of the Zn binding $\sim 5.5 \text{ \AA}$ from the Fe in a solution sample.

- 5) It is also not clear whether the scattering path used to fit the Fe-Zn distance is a single- or multiple scattering path (or a few of them).

Response: With regards to the scattering path used to fit the Fe-Zn contribution - a single scattering Fe-Zn path was used to model the Fe-Zn contribution. To ensure that this is clear, we have added "The Fe-Zn scattering interaction was modeled with a single Fe-Zn path" as a table footnote for Fig. 3 and Supplementary Fig. 5.

- 6) One comment to the estimated standard deviations for fitted distances: In the caption of Fig. 3f-g, the authors included the following comment: "aThe estimated standard deviations for distance are on the order of $\pm 0.02 \text{ \AA}$. bValues of σ^2 have been multiples by 105". I do not see how these standard deviations were estimated, especially that I would not expect a single bond length uncertainty DR to all paths included in fit results tables presented in Fig. 3g and Fig. S4b-c. My understanding is that depending on whether a given path is a single scattering or multiple scattering paths, the value of Debye-Waller factor σ^2 is a more adequate metric to say something about DR values. So, if I assume that Fe-N paths are single scattering paths, I can calculate that $\sigma^2 = 0.0021 \text{ \AA}^2$ and $\sigma^2 = 0.0034 \text{ \AA}^2$, for axial and equatorial Fe-N bond lengths fitted in Fig. 3g, respectively. This translates into $\sigma = 0.046 \text{ \AA}$ and $\sigma = 0.058 \text{ \AA}$, which is closer to 0.05 \AA and 0.06 \AA . Looking at other σ^2 , included in both Tables, the standard deviations will certainly be larger for longer scattering paths, especially for the distant Zn atoms. I would suggest correcting this statement or explain in case there is a justified way for this estimate.

Response: The estimates provided for the standard deviation of the bond lengths is standard for these types of data collected on beamlines 9-3 and 7-3 at SSRL (<https://doi.org/10.1021/jacs.3c01772>, <https://doi.org/10.1021/acs.biochem.6b00983>, <https://doi.org/10.1021/bi501489r>). However, we can also provide the fitting error, and have updated our fitting tables to include these values at the 1σ level. While the value of σ^2 is a measure of pair-wise disorder (representing both static and dynamic disorders) of a given component, we are unaware of its application in biological systems, as it is correlated with other parameters that make it less suitable as a tool to quantify disorder.

Reviewer #3 (Remarks to the Author):

In Grosjean et al., the authors report characterization of the a conserved protein domain capable of binding heme and perform a series of structural, phylogenetic, and mutational analyses in order to dissect the role of this protein domain. The authors are particularly focused upon a cyanobacterial protein (named Dri1) uniquely composed of the single conserved protein domain (DUF2470) without other evident protein domains. The authors detail a heme-dependent binding affinity between Dri1 and subunits of the succinate dehydrogenase complex.

As the expertise of this reviewer was sought in the area of cyanobacterial physiology and evaluation of photosynthetic activity, I will restrict my comments to the experiments, claims, and data interpretation associated with these topics. Unfortunately, there are serious concerns about the methodology used to analyze photosynthesis in the variety of Dri1 mutants that can considerably impact the conclusions that can be drawn.

Major comments:

The authors do not sufficiently describe the methodology behind the chlorophyll a fluorescence kinetics and the PQ redox state measurements at a level of detail to fully diagnose the data shown. However, the data appears to show some very unexpected results that may indicate some core methodological error(s). This reviewer suspects that this may be closely related to the author's use of a FluorCam 800MF: this line of instrumentation appears to have been developed for the analysis of plants. There are a number of substantial differences between the photosynthetic machinery between cyanobacteria and plants that require adjustment of the instrumentation and the equations used to evaluate cyanobacterial samples relative to plants when estimating photosynthetic performance. I elaborate upon a few concerns related to this analysis below:

Response: We appreciate the reviewer's time and thorough review of the this section of our manuscript, and given the concerns raised regarding the chlorophyll fluorescence kinetics, we repeated these experiments of chlorophyll fluorescence kinetics by collaborating with our colleagues Prof. Krishna Niyogi at UC Berkeley and Dr. Masakazu Iwai at Berkeley lab (two new co-authors) and used a Dual PAM-100 (Walz) as it is more standard in the field of photosynthesis in cyanobacteria (e.g., <https://link.springer.com/article/10.1007/s11120-017-0367-x>).

1. The authors don't fully describe how they are standardizing their cyanobacterial samples prior to Chl a fluorescence analysis. It is important to standardize the chlorophyll a content of samples during analysis in order to have a similar baseline. Additionally, because the phycobilins can contribute to F0 in cyanobacterial due to their overlapping excitation/emission spectra, when there is a large difference in F0 values seen (as in Figures 6AB), it may be important to do a more detailed pigment analysis to determine if the phycobilisome content or other reactive pigments are substantially altered between the different strains.

Response: We now further describe the standardization of the samples prior to Chl a fluorescence analysis (Chl a of $5 \mu\text{g mL}^{-1}$). It reads line 598 in the methods:

"WT parental strain and $\Delta dri1$ were grown in liquid BG-11 at 30°C with shaking at 90 rpm in an Innova® 44/44R shaker (New Brunswick). Log-phase cells were harvested and normalized to a chlorophyll concentration of $5 \mu\text{g mL}^{-1}$ (quantified spectroscopically after methanol extraction⁸⁹). One milliliter of chlorophyll-normalized cells were dark acclimated for 30 min and chlorophyll fluorescence measurements were performed on a Dual-PAM 100 (Waltz), to determine chlorophyll fluorescence parameters (F_0' , F_m' _{dark}) in the dark."

Furthermore, we included the absorption spectra analysis of the two strains (Fig. 6b), which demonstrates no changes in the pigment composition, supporting that the observed differences are not explained by a difference in pigment compositions between the two strains. We also made some changes in the text line 297. It now reads: "To account for any putative influence of differences in the pigment composition (e.g., highly fluorescent phycobilisomes) of the strains, we measured their absorbance spectra normalized to the optical density at 750 nm (Fig. 6b). No difference was observed between the two strains, suggesting that the higher fluorescence is not due to a modified pigment composition. This modified chlorophyll fluorescence was not the result of an altered accumulation of major photosynthetic membrane complexes (PSI and PSII) triggered by the absence of Dri1, as verified qualitatively

by blue native PAGE (BN-PAGE) (Fig. 6c). Calculation of the Q_A redox state (1-qP) at the end of the 5 min of actinic light (Fig. 6a) showed a higher value (0.81) in the $\Delta dri1$ mutant than in the WT (0.71). Therefore, the increased chlorophyll fluorescence displayed by $\Delta dri1$ implies a modified PQ redox state compared to the parental strain.”

2. Something is very unusual with the fluorescence trace provided in Figure 6b. Cyanobacteria are generally in State II (phycobilisomes associated predominantly with PSI) in the dark. Upon turning on actinic light, electrons are pulled from the PQ pool, leading to an overall oxidation of the photosynthetic electron transport chain and leading to a characteristic rise in Chla fluorescence as the cells shift to State I (PBS associated more with PSII). It is notable that this is a distinction between cyanobacterial analysis and plants. The authors show data that seems to describe the reverse (Chla fluorescence drops in the light).

a. This gives rise to some confusing data (for instance, the F_0 fluorescence is higher than the F'_0 fluorescence). Indeed, the F'_M (maximum fluorescence in the light) is substantially lower than F_0 (minimal fluorescence in the dark). Similarly the maximal fluorescence under a saturation pulse decreases as the cells are placed into light (as well as the variable fluorescence), a situation that is reversed from expected.

b. It is unclear how the authors are deriving their claims of increased F_v in the $Dri1$ mutant (claims beginning on line 261). Both the F_v in the dark, and the F'_v (variable fluorescence from F'_0 to F'_M) appear from the trace to be higher in WT. It is somewhat difficult to read these traces at the size given to the reviewers, but at all points along the WT trace (dark green) the size of the induced change in fluorescence via the saturation pulse appears to be greater than in the mutant trace. Regardless, it is somewhat difficult to interpret these results given that the state transition is likely still occurring throughout the entire 60 seconds of actinic light exposure at the actinic light intensity used.

c. While it is appreciated that the authors show the fluorescence across the well as a part of their data in Figures 6a and 6b, it is not easy to explain why there is a substantial difference in fluorescence observed across the sample, as these should be homogenous, liquid suspensions. Were any controls used to rule out artifacts of the use of the 96 well format (this is atypical) and were optically appropriate 96 well plates used?

Response: In addition to repeating the experiment with a Dual PAM-100, we also used longer time scales (5 min of actinic light instead of 1 min at which state transition likely still occurs) (Fig. 6a). The results obtained using the Dual PAM-100 are similar to typical fluorescence traces found in the literature for *Synechocystis* sp. PCC 6803 using orange actinic light (e.g., <https://link.springer.com/article/10.1007/s11120-017-0367-x>). In line with our previous results, we also observed an increased basal (F'_0), and maximal ($F'_{m\text{'dark}}$) chlorophyll fluorescence in the dark in the $\Delta dri1$ mutant compared to the WT strain (Fig. 6a), and our previous conclusions are not altered.

Chlorophyll fluorescence relies on the redox state of Q_A , the first electron-accepting PQ in PSII. Furthermore, in cyanobacteria, the redox state of the PQ pool is in equilibrium with Q_A . Therefore, an apparent chlorophyll fluorescence increase can result from an increased level of Q_A^- and can subsequently depict a more reduced PQ pool. Yet, other parameters can influence chlorophyll fluorescence, among which different pigments, with the example of phycobilisomes that are highly fluorescent. To account for any putative influence of differences in the pigment composition of the strains, as mentioned above, we measured their absorbance spectra normalized to the optical density at 750 nm. No difference was observed between the two strains, suggesting that the higher fluorescence is not due to a modified pigment composition (Fig. 6b). The composition of photosynthetic complexes is not different either as estimated by BN-PAGE (Fig. 6c). Therefore, the increased chlorophyll fluorescence F'_0 can be explained by an over-reduction of the plastoquinone pool, as has been reported several times in the literature (e.g., <https://doi.org/10.1016/j.bbaprot.2019.04.007>, <https://doi.org/10.1104/pp.16.00479>, <https://doi.org/10.1104/pp.20.00284>, <https://doi.org/10.3390/life11040279>, <https://doi.org/10.3390/life10050055>, <https://doi.org/10.1371/journal.pone.0139061>).

The results in line 289 now read: “We compared the chlorophyll fluorescence of these two strains as a proxy for any disruption or modification of components in the electron transfer chains. In cyanobacteria, the redox state of the plastoquinone (PQ) pool is in equilibrium with Q_A (the first electron-accepting PQ in PSII). Therefore, since chlorophyll fluorescence reflects the redox state of Q_A , it also indirectly reflects the redox state of the PQ pool⁴¹. Comparison of chlorophyll fluorescence kinetics revealed an increased minimum (F'_0) and maximum ($F'_{m\text{'dark}}$) chlorophyll fluorescence in the dark-acclimated state, as well as a higher steady-state fluorescence under actinic light (F_s) in $\Delta dri1$ compared to the WT (Fig. 6a). An apparent chlorophyll fluorescence increase could suggest an altered pigment composition, a possible impairment of PSII function, or an increased level of Q_A^- (i.e., a more reduced PQ pool⁴²⁻⁴⁴).”

3. Given the rather serious concerns with the measurements of some key parameters (e.g., F_0) in the above comments, it would be important to provide some additional information on the PQ reduction assay that was performed in panels 6D-6F and to elaborate upon controls. To the knowledge of this reviewer, this is an atypical experiment to perform on cyanobacterial samples.

Response: We have added more information on the assays performed in panels 6D-6F. Panel 6D is an analysis that has been extensively used for estimating the PQ pool redox kinetics in cyanobacteria, notably for the characterization of the function of the respiratory complexes NDH and SDH (see references <https://www.ncbi.nlm.nih.gov/pmc/articles/PMC94334/>, <https://www.ncbi.nlm.nih.gov/pmc/articles/PMC95315/> and <https://www.biorxiv.org/content/10.1101/2021.09.27.461999v1.full>). This assay was performed in the lab of our co-author Pr. Himadri B. Pakrasi who published previously the use of this assay to characterize various NDH mutants (<https://doi.org/10.1074/jbc.M003706200>). In this assay, the redox state of the PQ pool in thylakoids is monitored indirectly by determining the chlorophyll fluorescence yield in darkness upon addition of KCN, using a measuring light intensity low enough to not trigger any actinic effect. The fluorescence yield depends on the redox state of QA in PS II: the chlorophyll fluorescence yield is high when QA is reduced and low when QA is oxidized. In cyanobacteria, there is redox equilibrium between QA and PQ (<https://link.springer.com/article/10.1007/s11120-017-0367-x>). If oxidation of the PQ pool is blocked by the presence of potassium cyanide (KCN, inhibition of oxidases), and reduction by PSII is blocked by the absence of light (no activity of PSI and PSII), the reduction rate of the PQ pool by respiratory complexes therefore can be monitored qualitatively by measuring the chlorophyll fluorescence yield elicited by weak non-actinic pulses of measuring light.

In strains lacking SDH subunits (see references <https://www.ncbi.nlm.nih.gov/pmc/articles/PMC94334/>, <https://www.ncbi.nlm.nih.gov/pmc/articles/PMC95315/> and <https://www.biorxiv.org/content/10.1101/2021.09.27.461999v1.full>), the chlorophyll fluorescence is greatly reduced compared to the WT, reflecting an increase in the QA⁻ level due to PQ pool reduction. Deletion of *dri1* increased the reduction kinetics of the PQ pool relative to the WT strain as shown in Fig. 6d, supporting the chlorophyll fluorescence analysis results from Fig. 6a.

Some studies mention that the direct correlation between the redox state of the PQ pool and the intensity of the chlorophyll a fluorescence signal can be dependent on a range of physiological parameters (e.g., <https://doi.org/10.1104/pp.114.237313>). Therefore, we tested the Succinate:DCPIP oxidoreductase activity in isolated membranes from Fig. 6e and 6f, which refers to an assay that has been previously used to analyze the activity of SDH in cyanobacteria (see references <https://www.ncbi.nlm.nih.gov/pmc/articles/PMC94334/>, and <https://www.biorxiv.org/content/10.1101/2021.09.27.461999v1.full>).

Dichlorophenol indophenol (DCPIP, blue) has a maximum absorption at 600 nm. Upon reduction, DCPIP becomes colorless. Here we used DCPIP as an electron acceptor to inform on the oxidoreductase activity of cyanobacterial membranes, after addition of succinate, the substrate of succinate dehydrogenase. Controls without succinate were performed and subtracted, therefore the values reflect the succinate-dependent DCPIP oxidoreductase activity of the membranes. In our study (Fig. 6e-f) and the two references (<https://www.ncbi.nlm.nih.gov/pmc/articles/PMC94334/>, and <https://www.biorxiv.org/content/10.1101/2021.09.27.461999v1.full>), mutation of SdhB subunits decreases the DCPIP reduction by half of the WT strain. Mutation of *dri1*, however, increases the DCPIP reduction kinetics. This result is in line with the other complementary analyses we employed (calculation of the QA redox state (1-qP) at the end of the 5 min of actinic light (Fig. 6a), rise of fluorescence in the presence of KCN in darkness (Fig. 6d), and the succinate-dependent DCPIP oxidoreductase assay with isolated membranes (Fig. 6e-f)), which support the increased PQ pool reduction that is possibly the result of an increased SDH activity in the mutant Δ *dri1*.

Given these concerns, the conclusions reached related to Figure 6A-6F may be seriously compromised. There is a considerable amount of additional data in this manuscript that supports other elements of the conclusion and model that is ultimately proposed. However, the photosynthetic measurements (Lines 255-280) and the conclusions related to them (see Discussion beginning on Line 328) should be revisited.

Response: As mentioned above, we have repeated the photosynthetic measurements in collaboration with two experts known in the photosynthesis field. Given the new results, we have confidence in the conclusions we have drawn from these analyses. Additionally, the different analyses performed, including *in vivo* pull down assay, yeast two hybrid, structural modeling, mutagenesis studies on predicted interacting residues, chlorophyll fluorescence, PQ pool reduction and Succinate:DCPIP oxidoreductase activity in isolated membranes, fit the working model proposed where Dri1 is a negative regulator of SDH activity in *Synechocystis*.

Minor comment:

Elements of the language composition can be difficult to follow at times. For example, in the paragraph beginning on Line 221, there is considerable redundancy in the way that topic points are introduced and discussed. For instance, the section beginning on line 227 about SdhB1 repeatedly switches between HdrB and SdhC/D, repeats elements of the cyanobacterial SdhB1 c-terminal extension and repeats the cyanobacterial HdrB fusion. Streamlining the language may help reader comprehension. In some cases where the discussion of Sdh subunits more broadly (across many

prokaryotes and eukaryotes) is in the same discussion as the cyanobacterial-specific Sdh subunits, it may help to tweak the nomenclature so that it is clear what homolog(s) are being discussed.

Response: We have edited this paragraph and have gone through the MS to edit for language and clarity. This paragraph was reorganized, fleshed out and Fig. 5b and 5c were swapped. It now reads line 253:

“However, we were only able to reproduce the physical interaction between SdhB1 (sll1625) and Dri1 (Fig. 5a). Given the specific interaction between Dri1 and SdhB1, we performed a phylogenomic analysis of the iron-sulfur cluster binding subunit SdhB (Supplementary Fig. 13c-g) and a co-occurrence analysis between DRI, SDH subunits, and PNPOx-like domain (found in GBP and HugZ) throughout the tree of life (Fig. 5b). Through the co-occurrence analysis, we determined that while homologs of the four main SDH subunits, as described for *E. coli*³³, yeast³⁴, human³⁵ and plants³⁶ (the cytoplasmic flavoprotein subunit SdhA, the cytoplasmic iron-sulfur cluster binding SdhB, and the heme-binding membrane anchored subunits SdhC, and SdhD), generally exist in eukaryotes and bacteria, only two homologous subunits are found in cyanobacteria (SdhA and SdhB). While HdrB (also referred to as SdhE_E³⁷), appears to functionally substitute for SdhC and SdhD as characterized in *Campylobacter jejuni* and *Synechocystis*^{7,37,38} (Fig. 5b,c). In addition to substitution of SdhC/SdhD by HdrB, there are two SdhB homologs present in *Synechocystis*; sll1625 (SdhB1), which possesses the C-terminal extension conserved in all cyanobacteria, and sll0823 (henceforth denoted as SdhB2) (Fig. 5b, Supplementary Fig. 13b-d). Furthermore, co-occurrence analysis highlights the presence of SdhB1 and Dri1 orthologs strictly in cyanobacteria (Fig. 5b).”

Reviewer #4 (Remarks to the Author):

Response: We appreciate the reviewer’s review of the structural section of our manuscript. Given that both reviewer #4 and reviewer #5 focused on the structural side of the manuscript, amendments to the manuscript and responses to comments are complementary for the two reviewers.

1.How was the outer resolution limit determined, using CC1/2 or Mn(I)/sd? The authors might include CC1/2 in the crystallographic statistics table.

Response: Outer resolution limits were determined by STARANISO, CC1/2, I/sigI, Rmerge, and completeness. These details have been added to the Methods and the extended table of statistics previously listed in the Methods section is now merged with the Supplementary Tables 2 and 3.

2.Line 514-519: The authors described densities of the heme and disordered regions but did not provide any figures. It would be helpful if the authors include representative figures showing density maps around ligands and key residues to illustrate the quality of the maps.

Response: This comment is related to the first comment from Reviewer #5, therefore, for more details, please refer to our response to Reviewer #5 below. We have now included electron density maps (OMIT maps), for Dri1, Dri1 H21A and Dri1 H79A-R90A with heme to demonstrate the relevance and accuracy of the derived solved structures, with particular focus on the heme-binding site (heme, H79 and Zn ions) (Supplementary Fig. 3 and Supplementary Fig. 12). Furthermore, efforts were made throughout our study to employ independent and complementary analyses to strengthen our observations, conclusions, and hypotheses.

3.Some structures were refined with Refmac5 and others with Phenix. Was this selection random, or were there specific reasons for choosing these refinement programs?

Response: The refinement calculations were performed using two of the most utilized macromolecular refinement codes being Refmac5 and Phenix (<https://www.rcsb.org/stats/distribution-software>). The author contributions section now include details of which co-author worked on which structure, since the two co-authors preferred to employ different software. Generally speaking, both give broadly similar results. Nevertheless, to ensure that no artifacts or biases were unintentionally introduced, the Dri1 structures solved using Refmac5 were also solved using Phenix but did not improve the refinement statistics.

4.The gap between Rfree and Rwork can also be attributed to overfitting or an inappropriate resolution limit. Did the authors use different refinement strategies and resolution limits to obtain better Rfree and Rfree/Rwork gap values? Additionally, the real density map should be always inspected to avoid overfitting.

Response: In our case, and as mentioned in the methods section, the gap between Rwork and Rfree is the result of an effect of mild twinning. We used different refinement strategies to mitigate the gap between the R values. Twin law operators were applied, and the statistics presented here are the results of the best refinement strategy chosen for model refinement, by keeping the unbiased map as the reference. The gap values we obtained between Rfree and Rwork are in the accepted range by the community at this resolution (<https://www.ncbi.nlm.nih.gov/pmc/articles/PMC4420517/>, <https://www.ncbi.nlm.nih.gov/pmc/articles/PMC4282448/> and <https://www.medschool.lsuhsu.edu/biochemistry/docs/Evaluating%20X-ray%20crystal%20structure%20papers.pdf>).

5.All of the spectral figures, such as Supplementary Figure 8, are challenging to read. The colors used are too similar, and the curves appear very thin.

Response: We have modified the color scheme and width of lines used for Supplementary Fig. 8 to make it readable.

Reviewer #5 (Remarks to the Author):

Comments on "A hemoprotein with a zinc-mirror heme site ties heme availability to carbon metabolism in cyanobacteria".

The manuscript reports identification of a family of dimeric hemoproteins which is suggested to be a heme-binding regulatory domain. Authors suggest renaming of DUF2470 as Domain Related to Iron (DRI). DUF2470 entry in data bank describes it as an uncharacterised domain, found in a group of putative heme-iron utilisation protein. To substantiate this, they have focussed on a cyanobacterial specific protein clade where DRI is not fused to another domain which they have called Dri1 to distinguish it from DRI. Synthesis of heme and its regulation is important topics which makes this study of some interest and potentially worthy of its publication in Nature communications.

Authors have applied a number of experimental approaches as well as utilised computational modelling and predictive tools to support their hypotheses and characterise this interesting protein. Though a variety of techniques have been applied, some of the key data are of limited quality lacking robustness required for such new findings and publication in influential journals such as NCOMMS. As examples (but not limited to) I list some of the weaknesses:

1. There are four crystallographic structures, two for Dri1 (apo and apo+heme) and two for mutants (a single and a double mutation). Given the low molecular weight and a soluble nature of the protein, the data quality is poor, and resolution of the structures is what is considered these days very low. The best resolution is that of Dri1 (2.35Å) but with very high temperature factors. Even though the authors have not provided Wilson B for the data sets, I estimate that it is very high indeed reflecting poor quality of crystals calling for further optimization. Truncation of data may help but my guess is that it would not be sufficient. This comment is applicable to all of the 4 structures.

Response: Structures were solved from the best diffracting, single crystals. The data quality of each dataset was assessed by Rmerge, Rpim, I/sigI, completeness, and CC1/2 among others. The statistics are reasonable and are in the acceptable range as per the IUCr guidelines. Moreover, all structures have been deposited to the PDB and reviewed with no concerns. Twinning and non-crystallographic symmetry issues, discussed in the Methods section, could not be avoided despite many optimization attempts, impacting Rwork and Rfree values. High diffracting crystals were difficult to obtain due to some disorder and flexibility, particularly with non-covalently bound heme. As described in the Methods, samples were also sent for high-throughput screening to identify other crystallization conditions. These hits were extensively optimized to obtain the best diffracting crystal possible, and the resolution of the structures presented here is comparable to other solved structures of small proteins found in PDB. Moreover, the data we were able to collect may not have been successful without the technologies at NSLSII. In some instances, getting diffraction quality crystals from flexible smaller sized proteins is possibly even more challenging than for larger sized proteins. Furthermore, in a separate study focusing on the plant homolog of Dri1, which contains the DRI domain and a split-barrel domain and is 279 amino acids, we have achieved a structure at 1.8 Å resolution. Therefore, even among related proteins, there is no expectation that smaller proteins should be less challenging to solve. Despite these challenges, the quality and resolution of the data is sufficient to shed light on the heme binding and residue interactions.

We have now included electron density maps (OMIT maps), for Dri1, Dri1 H21A and Dri1 H79A-R90A with heme to demonstrate the relevance and accuracy of the derived solved structures, with particular focus on the heme-binding site (heme, H79 and Zn ions) (Supplementary Fig. 3 and Supplementary Fig. 12). Furthermore, efforts were made throughout the study to employ complementary experiments/analyses to strengthen our observations, conclusions, and hypotheses.

The reviewer also describes the B factors as very high where the range for acceptable B factors is acceptable given the lower resolution (for reference: <https://bmcbioinformatics.biomedcentral.com/articles/10.1186/s12859-018-2083-8>). That said, the reviewer does make a good point about high B factors for the ligand in the Dri1 H21A structure. Toward that, sentences were included explaining that heme is not bound tightly in Dri1 H21A and so is reflected by the higher B factor and the OMIT map. The loosely bound heme is also observed by MD-simulation (Supplementary Fig. 10). It now reads line 210;

“His21Ala exhibits a shifted binding interface with a rotated heme ring ($\sim 34^\circ$) relative to WT Dri1 (Fig. 4e-g, Supplementary Fig. 10). Calculated Polder OMIT maps with the heme, His residues, and bulk solvent excluded (Supplementary Fig. 12a) in addition to higher ligand B-factors (Supplementary Table 3) reflect a loosely bound heme.”

2. The combination of SEC-SAXS is good to see together with the latest approach which attempts to calculate electron density from a one-dimensional scattering curve. This approach is still not well-tested and not yet well known – thus a fuller description of strength and limitations should have been incorporated highlighting some of the key points from reference 61. It is important to remember that the electron density of proteins (~ 0.43 electrons/ \AA^3) is only slightly higher than the average electron density of water (0.33 electrons/ \AA^3) and any subtle changes in the buffer can make a significant difference. This is even more so important for smaller MW proteins. It is unclear in Figure 3(a) if they are of raw scattering curves (corrected for buffer) or smoothed. Curves in supplementary Figure 3 are of raw data but then fits to most of the curves are not particularly good. Non-uniqueness of the SAXS structures need to be taken into account when reporting part of the evidence for a significant advance.

Response: The reviewer makes a good point about the limitations of DENSS. Therefore, we have included additional information in the manuscript describing our use of the calculated electron density. As we were wary of overinterpreting the model, we used the density only as a visual guide to make sense of the data. In regards to the scattering curves, to help DENSS obtain more consistent fits, the SAXS profiles were rebinned to a nonuniform-step q -grid. This provided a small noise reduction at low q range but did not change the analysis and outcomes with SAXS-MD.

It now reads line 153:

“Electron density maps calculated by DENSS²⁶ yield consistent monomeric and dimeric reconstructions between datasets. To avoid overinterpretation^{27,28}, only the outermost contours (thresholds determined by approximate protein volumes) were considered in the comparisons, and scattering profiles were rebinned to a nonuniform-step q -grid to decrease noise in the low q region and aid with data fitting (Fig. 3a). Instead of relying on calculated electron densities, molecular dynamics (MD) ensembles were generated to model the possible solution state structures.”

And in methods line 479:

“Electron density maps were generated from averaging and refining 20 runs of DENSS²⁶ on scattering profiles rebinned to a nonuniform-step q -grid to reduce the noise in the low q range.”

The curves in Supplementary Fig. 4 and Supplementary Fig. 9 have been updated with longer time scales of SAXS-driven MD simulations from 12 ns to 30 ns. We also performed additional simulations with multiple starting structures to address the non-uniqueness of the SAXS structures and the models hence obtained fit better against the SAXS data as compared to the crystal structures (Supplementary Fig. 4c for Dri1 and Supplementary Fig. 9d and 9j for Dri1 H21A and Dri1 H79A-R90A, respectively).

3. Figure 4 caption – clarify that panels e to h are from experimental crystallographic structures. For panels c and d, SAXS raw data and fits should be shown.

Response: We updated the legend of Fig. 4 to specify that panels e to h are from experimental crystallographic structure. It now reads line 933: “**c-d**, SAXS-MD best fit model superimposed onto the crystal structures (cyan) of Dri1 His21Ala (**c**) and Dri1 His79Ala-Arg90Ala (**d**) and overlaid onto DENSS envelopes calculated from SAXS to $q_{\max} = 1.0 \text{ \AA}^{-1}$. Model to data fit for the SAXS-driven MD models and crystal structures are shown in Supplementary Fig. 6d and 6j, respectively. **e**, Superposition of dimeric structures obtained from crystallography - Dri1 (green), Dri1 His21Ala (light pink), and Dri1 His79Ala-Arg90Ala (purple). Respective hemes at the dimer interface are shown in stick representation. **f-h**, Selected side chains of Dri1 and variants involved in heme binding and interactions are shown (sticks) and labeled. Structures were determined from crystals.”

For panel c and d, the SAXS raw data and fits were updated and are now shown in Supplementary Fig. 9a-d for Dri1 H21A, and Supplementary Fig. 9g-j for Dri1 H79A-R90A.

4. There is very limited evidence (hard data) provided for the interaction between Dri1 and succinate dehydrogenase and the presence of a complex. At least a SAXS evidence for the complex in Figure 5 would have provided some confidence. Also, inability to identify a DRI fusion protein in a cyanobacterial is somewhat weakens the overall hypothesis.

Response: We absolutely agree that SAXS evidence for the complex would be great. Indeed, before preparing the manuscript, we attempted to do just that. Several attempts were made using different approaches. We made extensive attempts at the heterologous expression and purification of SdhB1 from *E. coli*, using two different C-terminal tags (His-tag and Streptag) and using different induction methods (IPTG induction and autoinduction media), and truncating the N/C termini to remove disordered/problematic regions. Given that SdhB1 is an iron-sulfur-cluster binding protein, we also tested the addition of precursors to alleviate the limited production of iron-sulfur-clusters that might be essential for the proper folding of the protein. We also attempted the co-purification of SdhB1 and Dri1 by combining the cell lysates of *E. coli* expressing either of the proteins in hopes they would stabilize each other, but this was also unsuccessful. These attempts are not quite surprising given the fact that every work that has been carried out on the SDH complex (or complex II in eukaryotes) was performed by purifying the entire complex and not individual subunits (e.g., <https://www.science.org/doi/10.1126/science.1079605>,

<https://www.sciencedirect.com/science/article/pii/S0005272801002298?via%3Dihub>,

<https://doi.org/10.1016/j.cell.2005.05.025>,

<https://onlinelibrary.wiley.com/doi/full/10.1111/tpj.14227>). However, in our case, purifying the native SDH complex from *Synechocystis* would not provide the information we are looking for, since all of our data point to a role for Dri1 has a protein that specifically binds to SdhB and inhibits complex formation. We were able to coelute SdhB1 and Dri1 from *Synechocystis*, but the purity and concentrations needed for immunodetection (presented in the MS) are orders of magnitude smaller than concentrations of highly pure complex required for SAXS studies.

Nevertheless, as stated previously, throughout the manuscript, we utilized diverse and complementary techniques and approaches, using *in silico*, *in vitro* and *in vivo* analyses. This was also the case for our description of the SdhB1-Dri1 interaction. As described in the manuscript, the hypothesis of this complex formation initially arose from a large-scale protein-protein interaction screen from *Synechocystis*. We confirmed this interaction with our own yeast-two-hybrid experimental data (Fig. 5a). We also modeled the complex *in silico* and identified residues that might be involved in complex formation (Fig. 5e-g), which we subsequently tested with yeast-two-hybrid experimental data providing evidence that not only do the proteins physically interact, but they interact in the manner that is predicted by the simulations (Fig. 5g-h). Finally, we confirmed their interaction *in vivo* in *Synechocystis* sp. PCC 6803 by protein pull-down assay (Fig. 5d), which allowed us to confirm the interaction with proteins expressed in the native host and test the role of heme in inhibiting the interaction. Furthermore, the reverse-genetic studies (Fig. 6) are also consistent with our working model of the interaction of these two proteins and the impact on succinate dehydrogenase complex activity and electron transfer to the plastoquinone pool of the electron transfer chain.

The reviewer mentions “Also, inability to identify a DRI fusion protein in a cyanobacterial somewhat weakens the overall hypothesis”. We do not agree nor necessarily understand how the absence of a DRI fusion in cyanobacteria would weaken our working hypothesis. Perhaps, this comment is derived from our hypothesis that when DRI is part of a fusion protein in other lineages, it is functioning as a regulatory domain. As such, as a regulator of succinate dehydrogenase, we may expect to see fusion proteins in some cyanobacterial genomes where DRI is fused to SdhB? While such a fusion protein would be fortuitous and provide another piece of bioinformatic data to support an interaction between Dri1 and SdhB, the vast majority of proteins that physically interact do not exist as fusion proteins based on large-scale comparative genomics analysis. Indeed, the best functionally equivalent homolog of Dri1 is GBP from the plant *Arabidopsis thaliana*. GBP is a fusion protein between DRI and a split-barrel domain, and it is a post-translational regulator of GluTR. It binds to GluTR to regulate it, but there are, like Dri1 and SdhB, no known fusion proteins between GluTR and GBP in any available plant genome.

5. The authors suggest an increase in heme-binding capacity as well as an increase in the heme-binding affinity with Zn. It would have been good to have crystallographic structures with and without Zn (but with heme) as it would have allowed to assess ‘any optimisation’ of heme stereochemistry at the homo-dimeric interface.

Response: We agree that an apo version of Dri1 +heme would have been ideal to obtain to answer this question. Indeed, before submission of the manuscript we performed extensive screens, but were unable to acquire crystals of the heme-bound Zn-minus form. As such, we instead characterized the apo variants in the solution state using XAS, SAXS, and EPR.

6. 20ns of MD calculations are on lower end of the scale. It is fairly simple to extend these to 100ns.

Response: As suggested by the reviewers, the free MD simulations were extended to 100ns and Supplementary Fig. 10 was updated accordingly. We have also included in Supplementary Fig. 11 the rmsd curves showing that the simulations are relatively stable past 20 ns.

3. Data quality of XAFS is good but some of the interpretation is unclear and may be over-stretched. It has not been made clear if the multiple scattering from the pyrrole rings have been taken into account. Figure 3 should also include a schematic of N-C paths that have been included and the multiple scattering (MS) contributions that have been included. For MS pathways, the angle information should also be included and if these were constrained or allowed to float. The assignments of peaks between 5 and 5.5Å should be handled with care.

Response: We have updated the manuscript and SI (Supplementary Fig. 5) to include this information by including a schematic of representative scattering paths from each multiple scattering interaction. Four sets of multiple scattering paths were incorporated in the fits, with distances (half-path lengths) of 1) 3.26 Å, 2) 4.32 Å, 3) 5.33 Å and 4) 5.70 Å. These paths are illustrated in Supplementary Fig. 5b. Several individual contributing paths were accounted for in each of these four fitted paths by increasing the coordination number to reflect the number of multiple scattering paths for a given distance. Obtaining the exact information on these would require higher quality EXAFS than we were able to obtain on these samples. As such, we disagree that angle information of the fit model can be provided. But we can provide the angle information of those paths present in the crystal structure that were used in fitting the EXAFS data. Representative angles are 1) 129°, 2) 165°, 3) 161°, 4) 145°. The multiple scattering paths each contained a σ^2 and ΔR that was allowed to float, given the constraints mentioned above (averaging of many paths and possibility of structural distortions in the heme as predicted in our MD simulations).

Scattering strength from such distance even for Zn is limited and their presence is largely in low k range. One way to gain some credibility would be to perform a difference EXAFS analysis i.e. subtracting the fit without Zn from the experimental spectra and then fitting the difference spectrum with Zn.

Response: Reviewer #2 had similar concerns. We have included a new figure (Supplementary Fig. 5d-e) in the manuscript justifying the inclusion of the Zn scattering. We have used Fourier filtering to show that the Zn EXAFS amplitude peaks at $k=10 \text{ \AA}^{-1}$ and has a significant contribution at high k. In contrast, the light atom contributions are at low-k and out of phase.

Current EXAFS fit involves too many parameters for the number of independent data points in the limited k-range.

Response: We have applied Stern's criteria and the number of Nind is 34.6 and the Nidp we used is 20-22 (depending on whether the first shell is modelled with one or two Fe-N paths).

The authors claim to have estimated standard deviations for distance on the order of $\pm 0.02 \text{ \AA}$ for all of the distances determined from EXAFS analysis. This is incorrect. They are fitting errors from an over-fitting situation – not the accuracy of distances. The first shell distance is accurate and is likely to be accurate to $\pm 0.02 \text{ \AA}$ but not others due to presence of multiple scattering, weaker contribution from distant atoms and overlapping contributions from outer shells that would result in cancellation of signals for part of the k-range. A thorough analysis of with the addition of each shell is required to assess the significance of contribution and the uncertainty in their determination.

Response: We agree with the reviewer and have changed our text to reflect the appropriate standard deviations obtained from fitting. We also agree with the reviewer that overlapping contributions from distant atoms and multiple scattering paths. In our new figure (Supplementary Fig. 5d-e), we have used Fourier-filtering to show what the scattering paths from these distant atoms are, and how they compare against the Fe-Zn contribution, illustrating that they cannot account for the peak in the Fourier-transformed EXAFS plot which we are designating as the Fe-Zn peak without the Zn contribution. We found that, when fitting shell-by-shell to assess how each additional path influences the distance uncertainties, as the reviewer requested, the uncertainty in the Fe-Zn path which results from the exclusion of the adjacent multiple scattering paths is less than the fit uncertainty (0.05 Å vs 0.06-7Å).

Minor points:

1. Authors need to clarify that heme binding and dimeric assembly can occur without Zn.

Response: We have clarified all instances of Dri1 bound with heme as Dri1 +heme. Additionally, apo-Dri1 with heme will be indicated as apo-Dri1 +heme. It now reads line 182: "Since Zn is not essential for heme binding (Supplementary Fig. 7), we compared apo-Dri1 with Zn-Dri1 and Co-Dri1 to better understand the role of zinc in heme binding. Zinc, but not cobalt, appears to significantly increase the heme-binding capacity and marginally increases the heme-binding affinity (Supplementary Fig. 6, Supplementary Fig. 7)."

2. Dri1 is mentioned in the abstract without specifying what it stands for.

Response: We modified the abstract to specify what Dri1 stands for. It now reads line 33; "we further demonstrate the existence of a functional link between heme binding by Dri1 (Domain related to iron 1, formerly *ssr1698*)"

3. Methods section is too long, some of it could be placed in supplementary section and some can be deleted altogether. Data collection and refinement statistics given between lines 548 and 549 are unnecessary and not normal. Given the supplementary tables 1 and 2, this information should be deleted from the methods section.

Response: As mentioned above in response to Reviewer #4, the extended table of statistics previously listed in the Methods section is now merged with the Supplementary Tables 2 and 3. In accordance with Nature Communication guidelines to authors, we transferred details and data analysis methods in Supplementary methods, resulting in a 50% decrease in the word count of the Methods section.

REVIEWER COMMENTS

Reviewer #1 (Remarks to the Author):

My technical comments have been mostly addressed. A few minor points are still missin though. Please fix these points. Further review is not needed.

- P- and T-coupling algorithms are not stated (such as Berendsen, Parrinello-Rahman, Nose-Hoover, velocity-rescaling). Also, the coupling time constant should be stated (MDP parameters tau-t and tau-p).

- Were all bonds constrained or only bonds involving hydrogen atoms? Using which algorithm? LINCS?

- Unit wrong? "the largest q point was set to 10 nm" -> nm⁽⁻¹⁾

- Missing word? "The SAXS-derived force was turned gradually" -> The SAXS-derived force was turned ON gradually.

- "particle-mesh Ewald method for long-range electrostatic interactions with a cutoff of 10 Å for non-bonded calculations": Should probably read: "... using a real-space cutoff of 10Å". Note that PME accounts for interactions also beyond the real-space cutoff, so the authors wording is somewhat misleading. In addition, please state explicitly the cutoff used for Lennard-Jones interactions.

Reviewer #3 (Remarks to the Author):

In their revised manuscript, the authors have made a serious and concerted effort to address the comments of my earlier review. These efforts include the recruitment of two experts in the field of photosynthesis and whom have considerable technical expertise in the area of chlorophyll a fluorescence analysis. Additional experiments were conducted on a fluorimeter that is routinely used for the analysis of cyanobacterial samples, and these data were used to replace prior analysis in the revised manuscript. Furthermore, sections of the main manuscript text and the supporting materials and methods appear to have been appropriately and substantively revised in accordance with these experiments.

I find that the revised manuscript is considerably easier to follow in the written sections related to photosynthetic analysis and now presents data in agreement with general expectations for cyanobacterial samples. Additionally, the data shown generally supports the conclusions drawn in relation to photosynthetic performance of cyanobacterial *dri1* mutants. While my prior comments focused mainly upon the photosynthetic parameters/analysis, I will reiterate here that other data presented in the manuscript is extensive and appears to support the author's conclusion of a novel heme-binding protein with functions in relation to succinate dehydrogenase.

Reviewer #4 (Remarks to the Author):

The authors have addressed all of my concerns.

Reviewer #5 (Remarks to the Author):

The comments I had made to the original manuscript were made in the spirit of constructive criticism with a view of improving the quality of the manuscript and the study. Authors have responded to these in a fiesty style rather than a response to my comments ensuring that appropriate adjustments to the presentation of results through text changes and changes in figures or their captions. For some of the comments, changes are made and others not. The ones where they need to pay attention rather than lecturing what is acceptable by IUCr or what are acceptable practices (as the reviewer is well aware of those), authors should act on them constructively putting possible limitations in the main text. Following are essential before acceptance should be considered:

1. Supplementary Tables 1 and 2 should be merged which now have Wilson B (as was assumed by me they are all very high).
2. Limited resolution and high Wilson B indicating high crystallographic disorder (which may be related to real disorder - dynamics or flexibility) should be explicitly mentioned in the main text before discussing the results. I would be content them saying at the end "Despite these challenges, the quality and resolution of the data is sufficient to shed light on the heme binding and residue interactions."

All other comments concerning, SEC-SAXS, MD and XAFS have been taken on-board constructively. Manuscript can be accepted if the above essential changes are made.

Reviewer #6 (Remarks to the Author):

Concerning the EXAFS fit procedures:

The recorded EXAFS is of very good quality and warrants the in-depth analysis undertaken.

The authors went into details identifying a possible Fe-Zn scattering path at 5.54 Å distance. I appreciate their efforts to try out possible other contributions including Fe-N/O at 5.54 Å and Fe-N-O multiple scattering paths, which yielded less agreement.

That said, if inclusion of the Fe-Zn scattering path is meant to argue that it can be present in the EXAFS measurement, then I am fine with this interpretation.

If it, however, is meant to actually prove its relevance in the recorded EXAFS, then I would appreciate including its statistical significance in a clearer way: For the reported R-factors I assume that these are always small, with or without the inclusion of the Zn path, since the bulk contributions of the first shell(s) are already dominant. I assume hereby the definition $R^2 = \frac{\sum ((\chi_{\text{cal}}(k) - \chi_{\text{exp}}(k))^2 / \chi_{\text{exp}}^2)}$, where the sum goes over every data point in their 2-13 Å⁻¹ range. I also see the inclusion of a $\chi\text{-red}^2$ value in their tables (in supplemental Fig. 5), which – to my understanding – refers to the definition $\chi\text{-red}^2 = 1/N\text{-data} \sum ((\chi_{\text{cal}}(k) - \chi_{\text{exp}}(k)) / \text{err})^2$, which thus delivers a value of the goodness of a fit with respect to its error bar size. Do the authors actually have the error bars for each data point? That would be splendid. Perfect fits would yield $\chi\text{-red}^2$ values of one, but the indicated values in their tables are much larger (on the order of 17). I am not sure, which definition the authors used in the present manuscript, but would suggest that – either way – the authors clarify their fit results by including the used formulas, since the outcome (whether or not the inclusion/exclusion of a Zn atom is significant) marks a prime point in their interpretation. This can be delivered in the SI.

As an added thought: if the authors would compare their data to FEFF calculations, where e.g., only the Fe-Zn distance is changed, e.g., in steps 0.1 Å and they plot their goodness of the fit as a function of this distance, one should observe a $\chi\text{-red}^2$ (or possibly R^2 , too) minimum right around their desired distance. I would consider this as a good indication that the authors indeed observed the Zn atom in their EXAFS study! Would this be possible?

We would like to once again thank all the reviewers for lending us their expertise and time to improve our manuscript with their thoughtful critiques. Below, we have pasted the reviewer comments and formatted them with italic text. Our response follows each comment in non-italicized text. Any new edits to the manuscript are pasted with line numbers corresponding to the track-changed version of the manuscript.

REVIEWER COMMENTS

Reviewer #1 (Remarks to the Author):

My technical comments have been mostly addressed. A few minor points are still missing though. Please fix these points. Further review is not needed.

Response: We are grateful for the reviewer's time spent reviewing our resubmission, their thorough review of our methods description and pointing out missing information that should be included. Their previous comments enabled us to substantially increase the quality of the manuscript, strengthen the conclusions of our study, and ensure that readers can more easily understand and use the methods described.

- P- and T-coupling algorithms are not stated (such as Berendsen, Parrinello-Rahman, Nose-Hoover, velocity-rescaling). Also, the coupling time constant should be stated (MDP parameters tau-t and tau-p).

Response: The T-coupling algorithm used was Nose-hoover with tau-t= 0.1ps while the P-coupling algorithm used was Berendsen with tau-p = 1.0ps. We have now added this information to the MS, it now reads lines 495-497:

"A 100 ns production simulation after energy minimization was conducted at constant temperature (300 K, using Nose-Hoover^{76,77} thermostat with a coupling time (tau-t) of 0.1 ps) and pressure (1 atm, using Berendsen barostat⁷⁸ with a coupling time (tau-p) of 1 ps)."

76. Nosé, S. A molecular dynamics method for simulations in the canonical ensemble. *Mol Phys* **52**, (1984).

77. Hoover, W. G. Canonical dynamics: Equilibrium phase-space distributions. *Phys Rev A (Coll Park)* **31**, (1985).

78. Berendsen, H. J. C., Postma, J. P. M., Van Gunsteren, W. F., Dinola, A. & Haak, J. R. Molecular dynamics with coupling to an external bath. *J Chem Phys* **81**, (1984).

- Were all bonds constrained or only bonds involving hydrogen atoms? Using which algorithm? LINCS?

Response: All bonds were constrained using the LINCS algorithm. We have now added this information to the MS, it now reads lines 501-502:

“LINCS algorithm⁷⁹ was used to constrain all bonds with an integration time step of 0.002 ps.”

79. Hess, B., Bekker, H., Berendsen, H. J. C. & Fraaije, J. G. E. M. LINCS: A Linear Constraint Solver for molecular simulations. *J Comput Chem* **18**, (1997).

- Unit wrong? "the largest q point was set to 10 nm" -> nm⁻¹

Response: That is correct, the unit should be nm⁻¹. We have made the correction to the text. It now reads lines 524-527:

“Briefly, the force constant (waxs-fc) was set to 1 for Bayesian inference, the memory time used (waxs-tau) was 250 ps, and 30 q points were coupled to the data where the largest q point was set to 10 nm⁻¹. The smallest q point was selected based on the experimental data for each case (i.e., 0.06 nm⁻¹ for Dri1 and His79Ala-Arg90Ala, 0.08 nm⁻¹ for His21Ala).”

- Missing word? "The SAXS-derived force was turned gradually" -> The SAXS-derived force was turned ON gradually.

Response: Yes, it should be the SAXS-derived force was turned on gradually. We have made the correction to the text. It now reads lines 527-528:

“The SAXS-derived force was turned on gradually by setting the waxs-t-target to 5 ns, with relative solvent uncertainty set to 0.1%.”

- "particle-mesh Ewald method for long-range electrostatic interactions with a cutoff of 10 Å for non-bonded calculations": Should probably read: "... using a real-space cutoff of 10Å". Note that PME accounts for interactions also beyond the real-space cutoff, so the authors wording is somewhat misleading. In addition, please state explicitly the cutoff used for Lennard-Jones interactions.

Response: A cutoff of 10 Å was set for Lennard-Jones interactions (i.e. short-ranged repulsive and dispersion interactions). We have made the changes to the text as suggested. It now reads lines 498-500:

“Lennard-Jones potentials were used to describe short-ranged repulsive and dispersion interactions with a cut off at 10 Å. The particle-mesh Ewald method⁸⁰ was used for non-bonded electrostatic calculations with a real space cutoff of 10 Å.”

80. Darden, T., York, D. & Pedersen, L. Particle mesh Ewald: An N · log(N) method for Ewald sums in large systems. *J Chem Phys* **98**, 10089–10092 (1993).

Reviewer #3 (Remarks to the Author):

In their revised manuscript, the authors have made a serious and concerted effort to address the comments of my earlier review. These efforts include the recruitment of two experts in the field of photosynthesis and whom have considerable technical expertise in the area of

chlorophyll a fluorescence analysis. Additional experiments were conducted on a fluorimeter that is routinely used for the analysis of cyanobacterial samples, and these data were used to replace prior analysis in the revised manuscript. Furthermore, sections of the main manuscript text and the supporting materials and methods appear to have been appropriately and substantively revised in accordance with these experiments.

I find that the revised manuscript is considerably easier to follow in the written sections related to photosynthetic analysis and now presents data in agreement with general expectations for cyanobacterial samples. Additionally, the data shown generally supports the conclusions drawn in relation to photosynthetic performance of cyanobacterial dri1 mutants. While my prior comments focused mainly upon the photosynthetic parameters/analysis, I will reiterate here that other data presented in the manuscript is extensive and appears to support the author's conclusion of a novel heme-binding protein with functions in relation to succinate dehydrogenase.

Response: We would like to thank the reviewer for taking their time to review our resubmission and are grateful for their positive and thorough assessment. Their previous comments enabled us to substantially increase the quality of our photosynthesis-related experiments and strengthen the conclusions of our study.

Reviewer #4 (Remarks to the Author):

The authors have addressed all of my concerns.

Response: We would like to thank the reviewer for taking their time to review our resubmission. Their previous comments enabled us to substantially increase the quality of and strengthen the conclusions of our study.

Reviewer #5 (Remarks to the Author):

The comments I had made to the original manuscript were made in the spirit of constructive criticism with a view of improving the quality of the manuscript and the study. Authors have responded to these in a fiesty style rather than a response to my comments ensuring that appropriate adjustments to the presentation of results through text changes and changes in figures or their captions. For some of the comments, changes are made and others not. The ones where they need to pay attention rather than lecturing what is acceptable by IUCr or what are acceptable practices (as the reviewer is well aware of those), authors should act on them constructively putting possible limitations in the main text. Following are essntial before acceptance should be considered:

- 1. Supplementary Tables 1 and 2 should be merged which now have Wilson B (as was assumed by me they are all very high).*
- 2. Limited resolution and high Wilson B indicating high crystallograpgic disorder (which may be related to real disorder - dynamics or flexibility) should be explicitly mentioned in the main text*

before discussing the results. I would be content them saying at the end "Despite these challenges, the quality and resolution of the data is sufficient to shed light on the heme binding and residue interactions."

All other comments concerning, SEC-SAXS, MD and XAFS have been taken on-board constructively. Manuscript can be accepted if the above essential changes are made.

Response: We are grateful to the reviewer for taking the time to critically review our resubmission and providing thoughtful and substantive comments. We sincerely apologize for responding to their previous comments and concerns in a feisty manner. That was not our intention. We genuinely appreciate their criticism and reminding us of the imperative to clearly convey limitations in the manuscript. Their previous and additional comments have enabled us to substantially increase the quality of the manuscript and strengthen the conclusions of our study. We have followed the reviewer's judicious guidance and have made the following edits to the manuscript.

- (1) We have combined the Supplementary Tables containing the data collection and refinement statistics for the reported crystal structures. Reference to Supplementary Tables in the main text and supplemental methods file have been renumbered accordingly.
- (2) We have now added the following text that highlights the limitations of the structures:
 - (a) Lines 131 to 134: "Because of potential structural flexibility and dynamics of the protein, we achieved a lower resolution with high Wilson B-factors (Supplementary Table 2) despite repeated efforts for data collection. Nonetheless, the quality and resolution of the data were sufficient to shed light on the heme binding and residue interactions."
 - (b) Lines 212 to 214: "Despite the lower resolution and crystal disorder (high Wilson B factors), the data are sufficient for elucidating heme-binding interactions."

Reviewer #6 (Remarks to the Author):

Concerning the EXAFS fit procedures:

The recorded EXAFS is of very good quality and warrants the in-depth analysis undertaken. The authors went into details identifying a possible Fe-Zn scattering path at 5.54 Å distance. I appreciate their efforts to try out possible other contributions including Fe-N/O at 5.54 Å and Fe-N-O multiple scattering paths, which yielded less agreement. That said, if inclusion of the Fe-Zn scattering path is meant to argue that it can be present in the EXAFS measurement, then I am fine with this interpretation.

Response: We are grateful for the reviewer's thoughtful critique of our data and conclusions. As the reviewer correctly assesses, our intentions with including the Fe-Zn scattering path in the Fe EXAFS of the WT is meant to argue that it can be present in the EXAFS measurement and is consistent with the presence of a Zn-Fe at a distance that agrees with the crystallography. To

make this more clear to the reader, we have changed the wording in the EXAFS results. It now reads lines 174 to 177:

“One Fe-Zn path can be fit to the Zn-Dri1 +heme dataset, which does not fit the apo-Dri1 +heme data. This Fe-Zn path at 5.54 Å is in reasonable agreement with the crystal structure, and is consistent with the existence of this zinc-mirror binding site (Fig. 3f-g, Supplementary Fig. 5a-c).”

If it, however, is meant to actually prove its relevance in the recorded EXAFS, then I would appreciate including its statistical significance in a clearer way: For the reported R-factors I assume that these are always small, with or without the inclusion of the Zn path, since the bulk contributions of the first shell(s) are already dominant. I assume hereby the definition $R^2 = \text{sum}((\text{chi-cal}(k) - \text{chi-exp}(k))^2 / \text{chi-exp}^2)$, where the sum goes over every data point in their 2-13 Å⁻¹ range. I also see the inclusion of a chi-red^2 value in their tables (in supplemental Fig. 5), which – to my understanding – refers to the definition $\text{chi-red}^2 = 1/N\text{-data} \text{sum}((\text{chi-cal}(k) - \text{chi-exp}(k))/\text{err})^2$, which thus delivers a value of the goodness of a fit with respect to its error bar size. Do the authors actually have the error bars for each data point? That would be splendid. Perfect fits would yield chi-red^2 values of one, but the indicated values in their tables are much larger (on the order of 17). I am not sure, which definition the authors used in the present manuscript, but would suggest that – either way – the authors clarify their fit results by including the used formulas, since the outcome (whether or not the inclusion/exclusion of a Zn atom is significant) marks a prime point in their interpretation. This can be delivered in the SI. As an added thought: if the authors would compare their data to FEFF calculations, where e.g., only the Fe-Zn distance is changed, e.g., in steps 0.1 Å and they plot their goodness of the fit as a function of this distance, one should observe a chi-red^2 (or possibly R^2 , too) minimum right around their desired distance. I would consider this as a good indication that the authors indeed observed the Zn atom in their EXAFS study! Would this be possible?

Response: Although it is not meant to actually prove its relevance in the recorded EXAFS, we have nevertheless responded to the reviewer's additional comments and have compared our data to FEFF calculations. For EXAFS analysis, the reduced χ^2 and r-factor equations are implemented in Artemis and can be found in this reference:

<https://doi.org/10.1002/9781118844243.ch11> (eq. 11.12 and 11.13) among others (for example, equation 45 in

https://millenia.cars.aps.anl.gov/xraylarch/downloads/2018Workshop/NewvilleEXAFS_RIMG78_ColorPreprint.pdf or in Scott Calvin's EXAFS for Everyone: <https://doi.org/10.1201/b14843>).

Unlike other uses of the reduced χ^2 statistic, its use in EXAFS analysis is in the form of $(\chi^2)/(\text{number of independent points} - \text{number of variables in the fit})$. Here the number of independent points is not the number of points in the spectrum (unfortunately), but a much smaller number estimated by the equation $N_{\text{idp}} = (2 \cdot \Delta k \Delta R) / \pi$ (sometimes with addition of +1 or +2). This is meant to be conservative, as the information content of an EXAFS spectrum is not ideally packed (<http://dx.doi.org/10.1103/PhysRevB.66.184303>). We note that our use of the r-factor and reduced χ^2 are the routine ones for EXAFS fitting. Artemis is a common platform for EXAFS analysis and is cited in our methods section. With respect to per-point error bars in the EXAFS spectrum, these are not typically reported, and we have not done so here.

With respect to the reviewer's request regarding moving the Fe-Zn distance around in 0.1 Å steps, we have done so in two ways. In the first, we froze all of the fitting parameters to those found in the best fit (this was done using the WT fit that has the first shell split 2+4), and then we varied the Fe-Zn distance. In the second approach, we froze the Fe-Zn distance and allowed the other parameters in the fit to optimize around the new Fe-Zn value. In both cases, we see a minimum at the Fe-Zn distance corresponding to the 5.54 Å observed in the paper. Below are four plots, for review only, demonstrating this trend.

All parameters floated, Fe-Zn fixed